# Generative Modeling of Full-Atom Protein Conformations using Latent Diffusion on Graph Embeddings

**Aditya Sengar**[*†‡]
aditya.sengar@epfl.ch

**Ali Hariri**[*]
ali.hariri@epfl.ch

**Daniel Probst**[§]
daniel.probst@wur.nl

**Patrick Barth**[†¶]
patrick.barth@epfl.ch

**Pierre Vandergheynst**[*‡]
pierre.vandergheynst@epfl.ch

## Abstract

Generating diverse, all-atom conformational ensembles of dynamic proteins such as G-protein-coupled receptors (GPCRs) is critical for understanding their function, yet most generative models simplify atomic detail or ignore conformational diversity altogether. We present latent diffusion for full protein generation (LD-FPG), a framework that constructs complete all-atom protein structures, including every side-chain heavy atom, directly from molecular dynamics (MD) trajectories. LD-FPG employs a Chebyshev graph neural network (ChebNet) to obtain low-dimensional latent embeddings of protein conformations, which are processed using three pooling strategies: blind, sequential and residue-based. A diffusion model trained on these latent representations generates new samples that a decoder, optionally regularized by dihedral-angle losses, maps back to Cartesian coordinates. Using D2R-MD, a $2\,\mu$s MD trajectory (12 000 frames) of the human dopamine D2 receptor in a membrane environment, the sequential and residue-based pooling strategies reproduce the reference ensemble with high structural fidelity (all-atom lDDT $\sim 0.7$; $C\alpha$-lDDT $\sim 0.8$) and recovers backbone and side-chain dihedral-angle distributions with a Jensen–Shannon divergence $< 0.03$ compared to the MD data. LD-FPG thereby offers a practical route to system-specific, all-atom ensemble generation for large proteins, providing a promising tool for structure-based therapeutic design on complex, dynamic targets. The D2R-MD dataset and our implementation are freely available to facilitate further research.

## 1 Introduction

Proteins function as dynamic molecular machines whose biological activities critically depend on transitioning between distinct conformational states [1, 2]. Landmark artificial intelligence methods like AlphaFold2 [3] and others [4–7] have advanced structure prediction but predominantly predict single static conformations, limiting their utility for systems with conformational heterogeneity. Accurate modeling of an ensemble of accessible conformations is essential to elucidate protein function and guide therapeutic design [8–11]. Crucially, these ensembles must explicitly represent all

---

[*]Signal Processing Laboratory (LTS2), EPFL, Lausanne, Switzerland

[†]Institute of Bioengineering, EPFL, Lausanne, Switzerland

[‡]Corresponding author

[§]Bioinformatics Group, University & Research Wageningen, The Netherlands

[¶]Ludwig Institute for Cancer Research, Lausanne, Switzerland

39th Conference on Neural Information Processing Systems (NeurIPS 2025).

atomic details, particularly side chains, whose subtle conformational rearrangements often govern molecular recognition and catalytic mechanisms [12].

Despite rapid advancements, existing generative models frequently fall short in capturing the detailed dynamics of side-chain movements specific to particular proteins [13]. Many powerful methodologies have focused either on *de novo* backbone designs [14–16] or on generating static all-atom structures [17, 18], but neither produce comprehensive conformational ensembles. Ensemble-generating approaches—via perturbations of static predictors [19, 20], flow-matched variants [21], or general MD-trained generators [22, 23]—typically operate at the backbone or coarse-grained level. Consequently, they fail to capture the intricate all-atom rearrangements, particularly involving side chains, that are critical for function. Although promising latent space [24–26] and physics-informed models [27, 28] have emerged, their capability to generate high-resolution all-atom ensembles from MD data reflecting functional transitions remains unproven, highlighting a significant unmet need for specialized generative frameworks.

G protein-coupled receptors (GPCRs) provide a compelling example of dynamic systems wherein precise all-atom modeling is indispensable [29]. This large family of transmembrane receptors, comprising over 800 human members, is responsible for mediating most known transmembrane signal transduction [30, 31] and is targeted by approximately 50% of all marketed drugs [32, 33]. GPCR signaling involves conformational transitions among multiple states, frequently induced by ligand binding [34], and occurs through intricate allosteric mechanisms where specific side-chain interactions are crucial [35, 36]. Capturing such dynamic, atomically detailed landscapes is vital to understanding receptor signaling specificity [37], biased agonism [38], and the design of drugs targeting unique allosteric sites [39–41], yet studying these events computationally remains formidable [42–44]. Current predictive methods for GPCR dynamics [45] do not generate the comprehensive all-atom conformational landscapes essential for mechanistic understanding.

To address this critical requirement, we introduce *Latent Diffusion for Full Protein Generation* (**LD-FPG**), a generative framework designed to **learn and generate diverse, all-atom conformational ensembles from existing MD simulation data of a target protein**, explicitly including side-chain details. Rather than simulating new trajectories, LD-FPG leverages extensive MD datasets. Our approach employs a Chebyshev Spectral Graph Convolutional Network (ChebNet) [46] to encode all-atom MD snapshots into a compact latent representation. A Denoising Diffusion Probabilistic Model (DDPM) [47] is then trained to explore this learned latent manifold, and latent representations are decoded back into full all-atom Cartesian coordinates. We demonstrate our framework on extensive MD simulations of the human dopamine D2 receptor (D2R), systematically evaluating distinct decoder pooling strategies. Our primary contributions in this paper are:

1. To the best of our knowledge, we present the first latent diffusion modeling framework specifically tailored to generate *complete, all-atom protein conformational ensembles*, capturing both backbone and **side-chain** dynamics directly from MD simulations.

2. We introduce and critically evaluate a novel graph-based autoencoder architecture utilizing ChebNet combined with distinct decoder pooling strategies, offering insights into dynamic protein ensemble generation.

3. Using the D2R system, we demonstrate our method's ability to generate high-fidelity ensembles, highlight the advantages of residue-based pooling, and assess the impact of auxiliary dihedral loss terms on generative accuracy.

Our approach provides a computationally efficient tool for exploring complex dynamics in switchable proteins, supporting both fundamental mechanistic studies and drug discovery applications. The remainder of this paper is structured as follows: Section 2 reviews related work; Section 3 details our proposed methodology; Section 4 presents the experimental setup and results; and Section 5 summarizes our findings and outlines future directions.

## 2   Related Work

**Generative Models for Protein Design and Static Structure Prediction.** Deep generative models have made significant strides in protein science, employing techniques such as diffusion [47], flow-matching [48, 49], and learned latent spaces [24, 50] to tackle complex structural tasks. For *de novo* backbone design, various methods, often inspired by the success of predictors like AlphaFold2

[3] and diffusion-based approaches like RFdiffusion [14] have emerged. Notable examples include FoldingDiff [15], alongside flow-matching models such as FoldFlow2 [49]. Latent space strategies have also been pivotal in this domain; for instance, LatentDiff [24] generates novel protein backbones using an equivariant diffusion model within a condensed latent space, while Ig-VAE [50] utilizes a variational autoencoder for class-specific backbone generation (e.g., for immunoglobulins). While powerful for creating new folds (e.g., Proteina [16]) or specific components, these methods typically target soluble proteins, often generate only backbone coordinates, and are not primarily designed for sampling multiple conformations of a specific, existing protein. The generation of complete static all-atom structures has also seen considerable progress. Models like Protpardelle [17] and Chroma [18] can produce full static structures from sequence information. Diffusion-based generative models such as AlphaFold3 [6] and Boltz-1 [7] also provide detailed single-state predictions of all-atom structures and complexes. Other approaches, like PLAID [25], integrate predictors with diffusion samplers. To complement backbone or static generation, methods including Flow-Packer [51] and SidechainDiff [52] focus on side-chain packing or prediction. However, these tools predominantly yield single static structures. Furthermore, decoupling backbone and side-chain generation risks overlooking their critical interplay during the complex dynamic transitions [29] relevant to the conformational landscapes our work aims to capture.

**Modeling Protein Conformational Diversity: From General Strategies to MD-Informed Approaches.** Beyond single static structures, capturing a protein's conformational diversity is crucial for understanding its function. Initial strategies to explore this diversity include learning from structural variations in experimental databases (e.g., Str2Str [20]) or perturbing static predictions to sample conformational space, as seen with AF2-RAVE [19] for GPCRs and AlphaFlow/ESMFlow [21] more broadly. While these methods effectively broaden sampling, capturing system-specific, native-like dynamics often benefits from models trained directly on simulation data, which can provide a richer representation of a given protein's accessible states. Among such simulation-informed approaches, latent space models have shown promise. For example, EnsembleVAE [53], trained on MD snapshots and crystal structures of the soluble protein K-Ras, generates $C_\alpha$ ensembles from sampled latent features with full-atom picture subsequently produced by RoseTTAFold [4]. Similarly, idpSAM [26], trained on extensive simulations of intrinsically disordered regions (IDRs) with an implicit solvent model, produces $C_\alpha$ trace ensembles that can then be converted to all-atom representations. Such approaches demonstrate the power of leveraging simulation data within generative frameworks, though their application has often focused on particular protein classes (e.g., soluble proteins, IDRs) or involved multi-stage processes for generating final all-atom structures. Further advancements in learning ensembles directly from MD simulations encompass a range of techniques. These include methods like ConfDiff [27] (force-guided diffusion), P2DFlow [54] (SE(3) flow matching), and MD-Gen [55] (continuous trajectories). Larger-scale models such as BioEmu [22] and Distributional Graphormer (DiG) [23] aim to learn from vast MD datasets or equilibrium distributions, while experimentally-guided approaches like EGDiff [28] integrate diverse data types. Collectively, these MD-informed methods signify substantial progress in modeling protein dynamics. However, a persistent challenge, particularly when these methods are applied generally or to very large datasets, is the consistent generation of high-resolution, all-atom ensembles that fully capture system-specific details. This is especially true for intricate side-chain rearrangements within native environments, such as lipid membranes for G protein-coupled receptors (GPCRs), which are essential for elucidating their functional transitions [29]. The development of generative models that can directly learn and sample such specific, all-atom conformational landscapes from relevant MD data, particularly for complex targets like GPCRs [44], thus remains an important frontier.

# 3 Methodology

The LD-FPG framework (Fig. 1) models an equilibrium ensemble as a distribution of *internal* deformations around a fixed, aligned reference $X_{\text{ref}}$. Rather than predicting absolute coordinates, we factor generation into two parts: (i) learning a score $\epsilon_\theta$ over a compact pooled latent that summarizes per-atom embeddings, and (ii) reconstructing full all-atom coordinates by decoding under a fixed reference context derived from $X_{\text{ref}}$. Training jointly optimizes a denoising objective in the pooled space and reconstruction losses (Cartesian with optional dihedral terms) on decoder outputs; at inference, individual conformational frames are sampled independently by drawing a latent vector from the learned distribution and decoding with the same reference context, without modeling temporal correlations between frames. Module specifics—encoder, pooling choices, diffusion sampler,

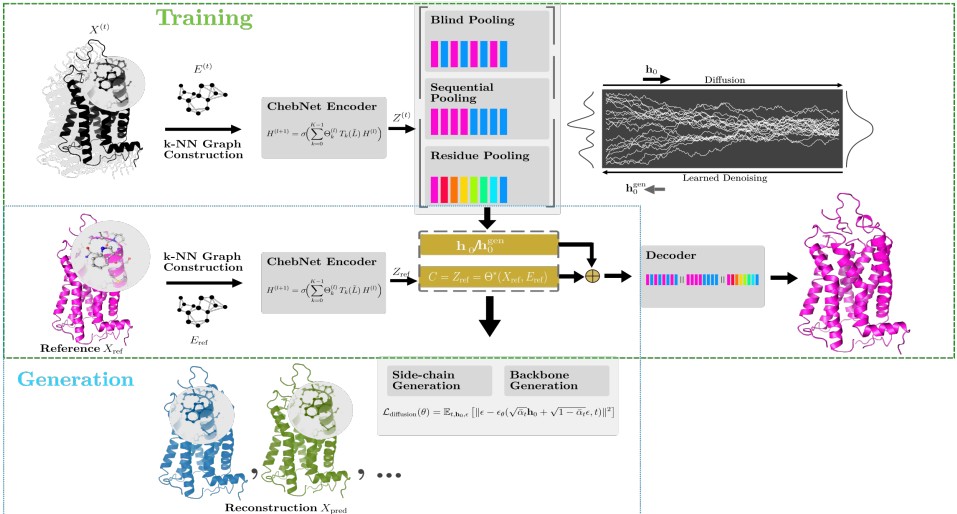

Figure 1: Overview of LD-FPG. Training (top, green): each MD frame $X^{(t)}$ and the fixed reference $X_{\text{ref}}$ are converted to $k$-NN graphs and encoded with a ChebNet into atom-wise embeddings $(Z^{(t)}, Z_{\text{ref}})$. A pooling module $\mathcal{P}$—chosen from blind, sequential, or residue-based strategies—compresses $Z^{(t)}$ to a low-dimensional latent $\mathbf{h}_0 = \mathcal{P}(Z^{(t)})$. A diffusion model is trained in this pooled space using a denoising loss to learn the score $\epsilon_\theta$ over $\mathbf{h}_0$. The decoder is trained concurrently as a reconstruction head: conditioned on the reference context $\mathbf{C} = Z_{\text{ref}}$, it maps $(\mathbf{h}_0, \mathbf{C})$ to full all-atom Cartesian coordinates $X_{\text{pred}}$, with losses on backbone and side chains (including optional dihedral-angle terms). Generation (bottom, blue): the trained diffusion sampler produces $\mathbf{h}_0^{\text{gen}}$, which, together with the same conditioning $\mathbf{C}$, is passed through the same decoder to yield an all-atom conformation $X_{\text{gen}}$. Frames are generated independently (no temporal conditioning).

decoder, conditioning, and loss functions—are detailed in Sections 3.2–3.5. Appendix A details the overall algorithm and notation; Appendices B–F provide component-level details, and Appendix J consolidates architectural and hyperparameter settings.

## 3.1 Input Representation and Preprocessing

Each MD snapshot $t$ is represented by a graph $G^{(t)} = (V, E^{(t)})$, where the node set $V$ comprises the $N$ heavy atoms and the node features are their 3D coordinates $X^{(t)} \in \mathbb{R}^{N \times 3}$. For each frame, the edge index $E^{(t)}$ is built on the fly by applying a $k$-Nearest Neighbors search to the aligned coordinates $X^{(t)}$ with $k = 4$. The node positions serve both as input features and regression targets. Prior to graph construction, the raw MD coordinates $\tilde{X}^{(t)}$ are rigid-body aligned to the first frame using the Kabsch algorithm [56] to remove global rotation and translation.

## 3.2 Latent learning of conformations

**Multi-hop encoding:** For each MD frame $t$, the encoder $\Theta$ maps the Kabsch-aligned heavy-atom coordinates $X^{(t)}$ and their $k$-NN graph $E^{(t)}$ to latent embeddings $Z^{(t)} \in \mathbb{R}^{N \times d_z}$, one $d_z$-dimensional vector per heavy atom. We implement $\Theta$ as a four-layer ChebNet [57] (see Appendix G.2 for architectural rationale) whose layers perform spectral graph convolutions with Chebyshev polynomials of order $K = 4$:

$$H^{(l+1)} = \sigma\Big(\sum_{k=0}^{K-1} \Theta_k^{(l)} \, T_k(\tilde{L}) \, H^{(l)}\Big), \tag{1}$$

where $H^{(l)} \in \mathbb{R}^{N \times F_l}$ are the node features at layer $l$ ($H^{(0)} = X^{(t)}$), $\tilde{L}$ is the scaled graph Laplacian, $T_k(\cdot)$ is the $k^{\text{th}}$ Chebyshev polynomial, $\Theta_k^{(l)} \in \mathbb{R}^{F_l \times F_{l+1}}$ are learnable weights, and $\sigma$ denotes a Leaky/ReLU non-linearity. Each layer is followed by BatchNorm, and the final output is

$L_2$-normalised per atom to yield $Z^{(t)}$. The embedding dimension $d_z$ was tuned; the best trade-off was obtained with $d_z{=}16$ for blind pooling, 8 for sequential pooling, and 4 for residue pooling.

**Conditioning Mechanism:** Each generated conformation is expressed as a deformation of a *reference* structure: the first Kabsch-aligned MD frame, $(X_{\text{ref}}, E_{\text{ref}})$. Rather than conditioning on raw Cartesian coordinates, we feed the decoder the reference's latent representation $C = Z_{\text{ref}} = \Theta^*(X_{\text{ref}}, E_{\text{ref}})$ where $\Theta^*$ denotes the frozen, pre-trained encoder parameters. This embedding compactly summarizes both 3-D geometry and graph topology, and in ablation studies outperformed using $X_{\text{ref}}$ directly. At generation time we copy $C$ to every sample in the batch, $C_{ex}$; the diffusion model then predicts only the atomic *displacements* from this common reference. This simplifies learning and guarantees that all sampled conformations stay anchored in the same chemical frame.

### 3.3 Decoder Architectures and Pooling Strategies

The decoder maps atom-wise latent embeddings $Z^{(t)} \in \mathbb{R}^{N \times d_z}$ (representing conformational deformations from $X_{\text{ref}}$, output by the encoder detailed in Section 3.2) and a conditioner $C \in \mathbb{R}^{N \times d_c}$ to all-atom coordinates $X_{\text{pred}} \in \mathbb{R}^{N \times 3}$. While these $Z^{(t)}$ embeddings are information-rich (as demonstrated in Section 4.2), their high dimensionality (e.g., up to $35k$ for D2R with $d_z = 16$) makes them computationally challenging for direct use as input to a diffusion model. Therefore, $Z^{(t)}$ is processed via a pooling strategy to yield a much more compact latent representation, $\mathbf{h}_0$ (typically $d_p \approx 60 - 100$ for Blind and Sequential strategies), which serves as the substrate for the diffusion model (Section 3.4). This compression is crucial, as preliminary experiments showed that $d_p > 200 - 300$ hampered diffusion training (for blind and sequential), while $d_p < 50$ degraded reconstruction quality. The efficacy of LD-FPG thus relies on the pooling strategy's ability to generate an informative yet compact $\mathbf{h}_0$. We investigate three strategies: Blind pooling, sequential pooling, and residue-based pooling.

**Blind pooling:** Atom-wise embeddings are globally pooled across all $N$ atoms using 2D adaptive average pooling $\mathcal{P}_{\text{global}}$ (reshaping $Z^{(t)}$ as an image-like tensor of size $N \times d_z$), yielding one context vector $z_{\text{global}} \in \mathbb{R}^{d_p}$ per sample in the batch (where $d_p = H \times W$ from the pooling dimensions). This global vector is tiled for each atom and concatenated with the corresponding broadcast conditioner vector $C^{(i)}$ to form the input $M_{\text{in}}^{(i)}$ for a shared MLP, $\text{MLP}_{\text{blind}}$, which predicts all atom coordinates $X_{\text{pred}}$ simultaneously.

**Sequential pooling:** Decoding is split into two stages. A *BackboneDecoder* first processes $Z^{(t)}$ and $C$ to output backbone coordinates $X_{\text{bb}}$. It typically pools backbone-specific embeddings to form a backbone context. Subsequently, a *SidechainDecoder* predicts side-chain coordinates $X_{\text{sc}}$ using $Z^{(t)}$, $C$, and the predicted $X_{\text{bb}}$. This stage often involves pooling sidechain-specific embeddings and combining this with backbone information and parts of the conditioner to form features for an MLP. The final structure is $X_{\text{pred}} = [X_{\text{bb}} \parallel X_{\text{sc}}]$. Three SidechainDecoder variants (arch-types 0–2) explore different feature constructions for the sidechain prediction MLP.

**Residue-based pooling:** This strategy models conformational changes as residue-level deformations relative to $X_{\text{ref}}$. For each residue $R_j$, its constituent atom embeddings $Z_{R_j}^{(t)}$ (a subset of the overall atom-wise deformations $Z^{(t)}$) represent its specific deformation from the reference state implicitly provided by $Z_{\text{ref}}$. These $Z_{R_j}^{(t)}$ are pooled via $\mathcal{P}_{\text{res}}$ into a local context vector $z_{\text{res},j} \in \mathbb{R}^{d_p}$, which summarizes residue $R_j$'s deformation. Each atom $i$ (in residue $R_{f(i)}$) then receives $z_{\text{res},f(i)}$ concatenated with its reference latent $C^{(i)}$ (from $Z_{\text{ref}}$) as input to $\text{MLP}_{\text{res}}$ for coordinate prediction. Thus, the decoder reconstructs atom positions from these summaries of residue-specific deformations relative to the reference.

### 3.4 Latent Diffusion Model for Generation

A Denoising Diffusion Probabilistic Model (DDPM) [47] operates on the pooled latent embeddings $\mathbf{h}_0$ derived from the chosen decoder pooling strategy. The model is trained to predict the noise $\epsilon$ that was added during a forward diffusion process. This training minimizes the standard DDPM loss function, $\mathcal{L}_{\text{diffusion}}$ (Eq. 2). New latent representations, $\mathbf{h}_0^{\text{gen}}$, are then sampled by iteratively applying the learned denoising network in a reverse diffusion process.

$$\mathcal{L}_{\text{diffusion}}(\theta) = \mathbb{E}_{t,\mathbf{h}_0,\epsilon} \left[ \| \epsilon - \epsilon_\theta(\sqrt{\bar{\alpha}_t}\mathbf{h}_0 + \sqrt{1 - \bar{\alpha}_t}\epsilon, t) \|^2 \right] \tag{2}$$

### 3.5 Loss Functions

The LD-FPG framework uses a series of Mean Squared Error (MSE) based losses, detailed in Appendix F, to train its components. The pre-training phase for the encoder $\Theta$ minimizes a coordinate reconstruction MSE ($\mathcal{L}_{\text{HNO}}$, Eq. 4). For all decoder strategies, the primary objective is coordinate accuracy, defined by the coordinate MSE loss $\mathcal{L}_{\text{coord}}$ (Eq. 5):

$$\mathcal{L}_{\text{Dec}} \approx \mathcal{L}_{\text{coord}} = \mathbb{E}_{(X_{\text{true}}, X_{\text{pred}})} \left[ \| X_{\text{pred}} - X_{\text{true}} \|_F^2 \right] \tag{3}$$

Specific loss configurations vary by decoder: **Blind pooling** begins with $\mathcal{L}_{\text{Dec}} = \mathcal{L}_{\text{coord}}$. If fine-tuned, it applies a weighted composite loss $\mathcal{L}_{\text{Dec}} = w_{\text{base}}\mathcal{L}_{\text{coord}} + S \cdot (\lambda_{\text{mse}}\mathcal{L}_{\text{mse\_dih}} + \lambda_{\text{div}}\mathcal{L}_{\text{div\_dih}})$, where $\mathcal{L}_{\text{mse\_dih}}$ (Eq. 6) and $\mathcal{L}_{\text{div\_dih}}$ (Eq. 7) represent dihedral angle MSE and distribution divergence terms respectively, applied stochastically (controlled by $S$) only for this strategy. **Residue-based Pooling** uses only coordinate MSE: $\mathcal{L}_{\text{Dec}} = \mathcal{L}_{\text{coord}}$. **Sequential Pooling** optimizes backbone ($X_{\text{predbb}}$) and full structure ($X_{\text{pred}}$) predictions in two stages via separate coordinate MSE losses $\mathcal{L}_{\text{BB}}$ and $\mathcal{L}_{\text{SC}}$, which correspond to applying $\mathcal{L}_{\text{coord}}$ to only backbone atoms or the full structure, respectively.

## 4 Experimental Setup

### 4.1 Experimental Setup

**D2 Receptor Dynamics dataset:** We perform extensive all-atom Molecular Dynamics (MD) simulations of the human Dopamine D2 receptor (D2R) to generate the input dataset. The system, comprising the D2R (2191 heavy atoms) embedded in a POPC membrane with water and ions, was simulated using GROMACS. The final dataset consists of 12,241 frames sampled every 100 ps after discarding initial equilibration. All frames were aligned to the first frame using the Kabsch algorithm. The data was then split into training (90%) and test (10%) sets. Static topology information, including atom indexing and dihedral angle definitions, was pre-processed. Further details on the simulation protocol, system preparation, and data pre-processing are provided in Appendix G.1 and Appendix B.

**Evaluation Metrics.** Model performance is assessed using metrics evaluating coordinate accuracy (e.g., MSE, lDDT, TM-score), dihedral angle distributional accuracy ($\sum$JSD), physical plausibility (steric clashes), and conformational landscape sampling (e.g., A100 activation index [58], PCA of latent embeddings). Auxiliary dihedral training losses are also reported where applicable for specific decoder configurations. Detailed definitions and calculation methods for all evaluation metrics are provided in Appendix G.3.

**Implementation Details.** All models were implemented in PyTorch [59] and trained using the Adam optimizer [60]. The overall three-phase training and generation workflow (encoder pre-training, decoder training, and diffusion model training) is detailed in Appendix A (Algorithm 1).

### 4.2 Results and Analysis

We evaluated LD-FPG's ability to generate high-fidelity, all-atom protein conformational ensembles via a multi-stage analysis: (1) assessing ChebNet encoder quality to establish an upper fidelity bound; (2) analyzing decoder reconstruction from ground-truth latents to isolate decoder errors; and (3) evaluating conformational ensembles from the full latent diffusion pipeline. Model lDDT scores (vs. $X_{\text{ref}}$) are interpreted against the "Ground Truth (MD) Ref" lDDT (Tables 1, Appendix H), which reflects the MD ensemble's average internal diversity relative to $X_{\text{ref}}$. Scores near this MD benchmark suggest a good balance of structural fidelity and diversity; significantly lower scores imply poorer fidelity, while substantially higher scores (approaching 1.0) might indicate insufficient diversity and over-similarity to $X_{\text{ref}}$.

**Multi-hop encoding fidelity of ChebNet** The ChebNet encoder generates high-fidelity atom-wise latent representations ($Z^{(t)}$) directly from input conformations. Reconstruction from these unpooled embeddings is excellent (details in Appendix H): for the D2R system ($N = 2191$ atoms), an encoder

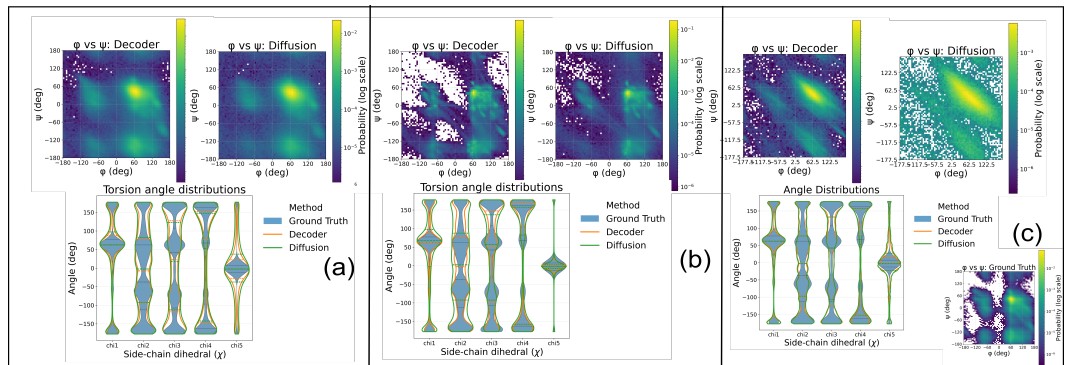

Figure 2: Comparison of dihedral angle distributions for different pooling strategies. (Top row) Ramachandran plots ($\phi$ vs $\psi$, log probability) comparing Decoder Reconstruction (left) and Diffusion Generation (right) outputs. (Bottom row) Violin plots comparing 1D sidechain dihedral angle ($\chi_1-\chi_5$) distributions for Ground Truth (MD, blue), Decoder Reconstruction (orange), and Diffusion Generation (green). (a) Blind pooling. (b) Sequential pooling. (c) Residue pooling.

configuration with a latent dimension $d_z = 16$ achieves a backbone MSE ($\text{MSE}_{\text{bb}}$) of $0.0008$ and dihedral JSDs around $0.00016$. This high-dimensional $Z^{(t)}$ (up to $2191 \times 16 = 35,056$ dimensions in this case) establishes a strong upper benchmark for the atomic detail initially captured. For efficient downstream diffusion modeling, these rich $Z^{(t)}$ embeddings are significantly compressed via a pooling step (Section 3.3) into a much more compact latent vector, $\mathbf{h}_0$. This pooling focuses the learning process on conformational *deformations* relative to a reference structure ($X_{\text{ref}}$, provided to the decoder via $Z_{\text{ref}}$), rather than encoding the entire static fold. While this necessary compression means the final generative pipeline's coordinate accuracy may not fully match the encoder's standalone reconstruction capabilities, the high intrinsic fidelity of the initial $Z^{(t)}$ ensures that $\mathbf{h}_0$ is distilled from a robust, deformation-rich representation.

**Decoder Reconstruction Fidelity** We next assessed decoder reconstruction of all-atom coordinates from these ground-truth ChebNet latents ($Z^{(t)}$), isolating decoder-specific errors (Table 1). The **blind pooling** decoder (using $d_z = 16$ atom features) achieved good coordinate accuracy ($\text{lDDT}_{\text{All}}$ $0.714$) but had limited dihedral precision, evidenced by a blurred Ramachandran plot (visualizing backbone $\phi, \psi$ angles) and smoothed $\chi$-angle distributions ($\sum\text{JSD}_{\text{sc}}$ $0.0290$; Figure 2a, orange traces/distributions). Optional dihedral fine-tuning yielded minimal JSD improvement while slightly reducing lDDTs. In contrast, **sequential pooling** (from $d_z = 8$ atom features) yielded excellent coordinate accuracy ($\text{lDDT}_{\text{All}}$ $0.718$) and superior backbone geometry, marked by sharp Ramachandran plots (Figure 2b, orange trace) and the lowest backbone dihedral divergence ($\sum\text{JSD}_{\text{bb}}$ $0.0026$). Its sidechain $\chi$-angle distributions were also well-reproduced ($\sum\text{JSD}_{\text{sc}}$ $0.0192$; Figure 2b, orange distributions). Intriguingly, **residue pooling** (from $d_z = 4$ atom features) excelled locally, achieving the lowest backbone/sidechain MSEs ($0.083/0.2257$) and the best sidechain distributional fidelity ($\sum\text{JSD}_{\text{sc}} = 0.0125$) with outstanding $\chi$-angle reproduction (Figure 2c, orange distributions). This local strength, despite a broader global backbone dihedral distribution ($\sum\text{JSD}_{\text{bb}}$ $0.0078$) and a "hazier" Ramachandran plot (Figure 2c, orange trace) partly due to its smaller per-atom latent di-

Table 1: Decoder Reconstruction Performance

| Decoder Configuration | $\text{lDDT}_{\text{All}}$ ↑ | $\text{lDDT}_{\text{BB}}$ ↑ | $\text{TM}_{\text{All}}$ ↑ | $\sum\text{JSD}_{\text{bb}}$ ↓ | $\sum\text{JSD}_{\text{sc}}$ ↓ | $\text{MSE}_{\text{bb}}$ ↓ | $\text{MSE}_{\text{sc}}$ ↓ | $\sum\mathcal{L}_{\text{dih}}$ MSE ↓ |
|---|---|---|---|---|---|---|---|---|
| Blind pooling ($d_z = 16$) | 0.714 | 0.792 | 0.961 | 0.0032 | 0.0290 | 0.1102 | 0.3934 | 0.3802 |
| + Dih. Fine-tuning | 0.698 | 0.776 | 0.960 | 0.0029 | 0.0279 | 0.0971 | 0.3564 | 0.2849 |
| Seq. pooling ($d_z = 8$) | 0.718 | 0.800 | 0.961 | 0.0026 | 0.0192 | 0.1291 | 0.5130 | 0.5164 |
| Residue pooling ($d_z = 4$) | 0.704 | 0.777 | 0.962 | 0.0078 | 0.0125 | 0.083 | 0.2257 | 0.2163 |
| Ground Truth (MD) Ref | 0.698 | 0.779 | 0.959 | - | - | - | - | - |

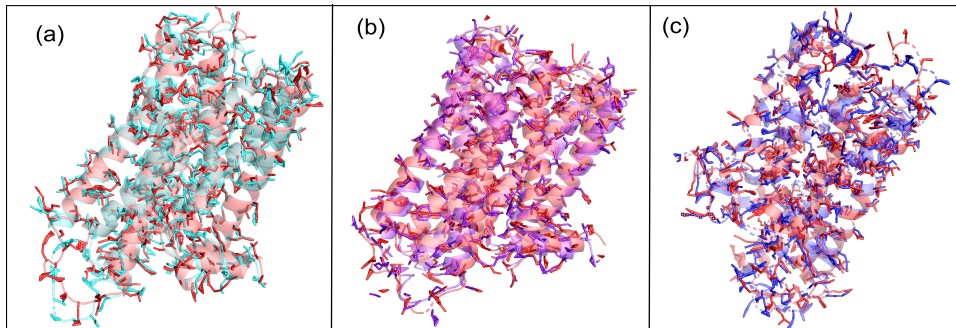

Figure 3: Examples of generated D2R conformations using different pooling strategies. Panels likely correspond to: (a) Blind pooling, (b) sequential pooling, and (c) residue pooling. Structures are visualized to show overall fold and sidechain placement.

mension, stems from its architecture. Pooling features within each of D2R's $N_{res} = 273$ residues into $d_p = 4$ local contexts ($z_{res,j}$) provides the MLP with access to a rich information space (effectively $N_{res} \times d_p \approx 1.1k$ dimensions describing overall residue deformations), boosting local performance. The quality of this decoder stage is key, as the diffusion model samples the distribution defined by these pooled latent embeddings $\mathbf{h}_0$.

Table 2: Diffusion Generation Performance.

| Model Configuration | lDDT$_{All}$ ↑ | lDDT$_{BB}$ ↑ | TM$_{All}$ ↑ | $\sum$JSD$_{bb}$ ↓ | $\sum$JSD$_{sc}$ ↓ | Avg. Clashes ↓ |
|---|---|---|---|---|---|---|
| Blind pooling | 0.719 | 0.792 | 0.964 | 0.006582 | 0.04185 | 1350.5 |
| + Dih. Fine-tuning | 0.683 | 0.748 | 0.9321 | 0.00648 | 0.0409 | 1340.9 |
| Sequential pooling | 0.712 | 0.801 | 0.942 | 0.0029 | 0.02895 | 1220.5 |
| Residue pooling | 0.6880 | 0.7575 | 0.9570 | 0.0117 | 0.0224 | 1145.6 |

**Diffusion Generation Quality** The ultimate test is the quality of conformational ensembles from the full LD-FPG pipeline (Table 2), where pooling strategies yield distinct ensemble characteristics. **Blind pooling** produces structures with the highest global coordinate accuracy scores (lDDT$_{All}$ 0.719, TM$_{All}$ 0.964). However, this global fidelity, likely impacted by the aggressive compression to its final latent $\mathbf{h}_0$, sacrifices finer details: its Ramachandran trace (Figure 2a, green trace) is somewhat blurred, side-chain distributions are over-smoothed ($\sum$JSD$_{sc}$ 0.04185), notably higher than

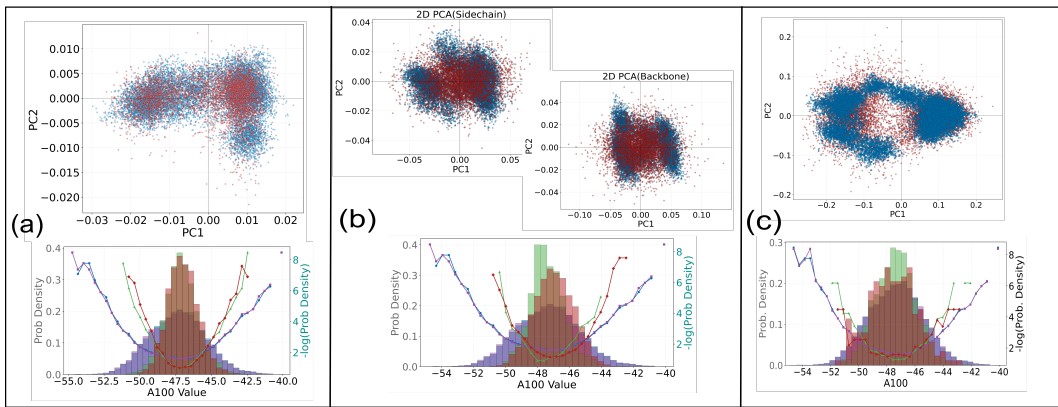

Figure 4: Latent space and collective variable (A100) analysis for Blind (a), Sequential (b), and Residue (c) pooling. Top rows: 2D PCA of dataset (blue) vs. diffusion-generated (red) pooled embeddings (Sequential shows separate backbone/sidechain PCAs). Bottom rows: A100 Value distributions (histograms) and PMFs (-log(Probability Density)) for dataset: ground truth (blue), autoencoder (purple), decoder (green), and generated ensembles (red).

the $\approx 1023$ clashes averaged by the ground truth MD ensemble. **It is crucial to note that dynamic MD ensembles naturally exhibit transient steric clashes, making the MD simulation itself, rather than an idealized zero-clash structure, the appropriate physical baseline for evaluating plausibility.** Rendered structures (Figure 3a) show well-placed helices but wandering $\chi$-angles, leading to protruding or clashing side chains. **Sequential pooling** offers an excellent all-round compromise. It maintains sharp backbone geometry (lowest $\sum\text{JSD}_{\text{bb}}$ 0.0029; clean Ramachandran in Figure 2b, green trace), respectable global scores ($\text{lDDT}_{\text{All}}$ 0.712), and good side-chain realism ($\sum\text{JSD}_{\text{sc}}$ 0.02895), with an average of 1220.5 clashes. Visually (Figure 3b), models are largely tidy. However, some surface side-chains may still flare out or collide. This aligns with its $\sum\text{JSD}_{\text{sc}}$ being better than blind but not residue pooling, suggesting some $\chi$-angles outside dense rotamer clouds, possibly due to the side-chain decoder's pooled backbone context leading to ambiguity that diffusion "averages". Thus, while delivering a sharp backbone and strong side-chain stats, a few rotamers (preferred side-chain conformations) may remain in incorrect basins. Its global backbone statistics are lower after diffusion ($\text{lDDT}_{\text{All}}$ 0.6880; $\sum\text{JSD}_{\text{bb}}$ 0.0117; Figure 2c, green trace), with occasional backbone kinks (Figure 3c), partly due to its use of a smaller ($d_z = 4$) per-atom latent dimension. However, its per-residue focus excels at $\chi$-angle distributions (best $\sum\text{JSD}_{\text{sc}}$ 0.0224) and yields the lowest average clash count (1145.6, closest to the MD reference), producing the tightest side-chain packing (Figure 3c). In essence, Figure 2 (dihedrals) and Figure 3 (structures) reveal these strategies' distinct behaviors: blind pooling prioritizes global fold over side-chain details; sequential pooling balances these, with minor outliers; residue pooling excels at rotamers, sometimes with less regular global backbones.

Figure 4 further explores conformational landscape coverage using PCA of latent embeddings and the A100 collective variable, an index of D2R activation states that proxies the MD-sampled landscape. It provides a dual analysis: PCA plots (top row) compare diffusion-generated latent distributions ($\mathbf{h}_0^{\text{gen}}$, red) against MD-derived ones (blue), while A100 value distributions (bottom row) assess the replication of the MD conformational landscape (and its autoencoder/decoder capture) by the full diffusion pipeline. The diffusion models' ability to cover major PCA regions of the MD latents positively indicates effective learning of the underlying data distribution. When examining the A100 distributions, Blind and Sequential Pooling, operating on more compact, globally-pooled latent spaces ($\mathbf{h}_0$ with $d_p \approx 100$), show reasonable landscape coverage, with Sequential closely tracing the MD distribution. However, **residue pooling** (Figure 4c, red curve) achieves the most comprehensive A100 landscape capture, populating both primary and subtler MD-observed states often missed by other methods. This superior recovery, despite moderate global backbone metrics, stems from its distinct latent space configuration. Unlike the compact, global latents of Blind/Sequential strategies, residue pooling's diffusion model leverages a significantly larger effective latent space formed by all per-residue contexts ($N_{\text{res}} \times d_p \approx 1.1k$ for D2R with $d_p = 4$). This richer, structured space, even if derived from a smaller $d_z = 4$ per-atom encoding, enables more nuanced representation of conformational substates. Its proficiency in local rotamer modeling is likely a key contributor, as accurate side-chain placement is crucial for the subtle cooperative shifts defining A100. The decoder's A100 distribution for residue pooling (Figure 4c, green curve) already indicates robust MD landscape coverage, which the diffusion model effectively samples. Visualizations for residue pooling in Figure 4 employed multi-epoch sampling—aggregating samples from checkpoints across different DDPM training stages, Appendix I.

### 4.3 Benchmark

To contextualize LD–FPG, we benchmarked against leading ensemble generators applicable to proteins—AlphaFlow [21], BioEmu [22], and Boltz–2 (MD–conditioning) [7]. We evaluate on the D2R system using two complementary axes: (i) *local geometric accuracy*, via the Jensen–Shannon divergence (JSD) of backbone dihedrals ($\phi, \psi$), and (ii) *dynamic flexibility*, via the mean C$\alpha$ root-mean-square fluctuation (RMSF), which quantifies ensemble diversity.

**Superior local geometry.** LD–FPG attains a backbone dihedral JSD of 0.007, improving over baselines by 3–5$\times$, indicating state-of-the-art recovery of local conformational preferences. **Flexibility vs. static fidelity.** BioEmu and Boltz–2 produce high-fidelity static structures (lDDT $\approx 1.0$) but exhibit very low ensemble flexibility (RMSF 0.07–0.09 nm). AlphaFlow shows higher flexibility (RMSF $\approx 0.84$ nm) yet still under-represents the ground-truth dynamics. **LD–FPG matches MD-level flexibility** (RMSF 1.22 nm vs. MD 1.34 nm) while preserving superior local geometry. Consequently, its lDDT ($\sim 0.80$) is lower than static predictors by design: for an ensemble generator,

Table 3: Benchmark comparison of LD-FPG against state-of-the-art models on the D2R ensemble. LD-FPG's lDDT is lower as it models the full, flexible ensemble, whereas baselines produce overly rigid structures, resulting in artificially high lDDT against the static reference. All baseline metrics were recomputed using their publicly available code. The Ground Truth MD (Ref) row indicates the properties of the original simulation data.

| Model | Backbone JSD $(\phi, \psi) \downarrow$ | Backbone lDDT | Backbone TM-score | Backbone RMSF (nm) $\leftrightarrow$ |
|---|---|---|---|---|
| LD-FPG (Ours) | **0.007** | $\sim$0.80 | $\sim$0.96 | **1.22** |
| BioEmu | $\sim$0.022 | 0.999 | 0.925 | 0.09 |
| AlphaFlow | $\sim$0.023 | 0.859 | 0.993 | 0.84 |
| Boltz-2 (MD-cond) | $\sim$0.034 | 0.997 | 0.975 | 0.07 |
| Ground Truth MD (Ref) | (Ref) | (Ref) | (Ref) | 1.34 |

lDDT to a single reference frame decreases as realistic conformational spread increases. **All-atom advantage.** Unlike backbone-centric baselines, LD–FPG generates full side-chain detail, enabling side-chain validation: we obtain low summed JSD over $\chi$ angles ($\sim$ 0.022) and all-atom RMSF (1.36 nm) close to MD (1.60 nm).

## 5 Conclusion and Future Work

We introduced LD-FPG, a latent diffusion framework generating all-atom protein conformational ensembles from MD data, demonstrated on D2R GPCR. Crucially, benchmarking showed LD-FPG uniquely matches MD-level dynamic flexibility (RMSF) while achieving superior local geometry, unlike more rigid baseline methods. It captures system-specific dynamics, including side-chain details, via learned deformations from a reference. Blind pooling offered global fidelity but compromised side-chain detail and clashes. Sequential pooling provided a strong balance, especially for backbone geometry. Residue pooling excelled in local side-chain accuracy and landscape coverage, despite some global backbone trade-offs and needing multi-epoch sampling for full diversity.

Future work includes targeted enhancements. For **Blind and Sequential pooling**, exploring larger pooled latent dimensions ($d_p$) is immediate, potentially improving detail capture but needing more extensive, diverse training data (e.g., from multiple related protein systems). For **Residue pooling**, with its promising high-dimensional effective latent space ($N_{\text{res}} \times d_p$), better denoisers are key to overcome multi-epoch sampling limits. This might use complex MLP/convolutional architectures, attention for its structured latent vector, or alternative generative models (score-based, flow-matching). For **Sequential pooling**, improving denoiser coverage of the MD-derived latent space (Fig.4b, backbone/sidechain overlap) could resolve side-chain misplacements and boost performance. Broader initiatives will enhance physical realism and architectural sophistication by incorporating lightweight energy surrogates, physics-guided diffusion schemes [61, 62, 28], and advanced architectures like attention-based fusion mechanisms [63–65], SE(3)-equivariant GNNs/transformers, or flow-matching generators [48, 66]. A long-term goal is generalizing LD-FPG by training on data from multiple related proteins (e.g., Class A GPCRs in various states) for foundational models akin to specialized "LLMs for protein dynamics." Efforts will also cover tailoring pooling for applications, scaling to larger systems [18], and rigorous benchmarking.

**Societal Impact**: LD-FPG can offer benefits by accelerating drug discovery, improving biological understanding, and enabling protein engineering for medicine/biotechnology. Ethical issues include dual-use, equitable access, data privacy for sensitive data, and rigorous validation to prevent misdirected efforts.

## Acknowledgments

A.S. was supported by a grant from the Center of Intelligence Systems (EPFL) to P.B. and P.V. This work was also supported by the Institute of Bioengineering (EPFL), the Signal Processing Laboratory (LTS2, EPFL), the Ludwig Institute for Cancer Research, and the Swiss National Science Foundation (grants 31003A_182263 and 310030_208179).

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

# A  Methodology Overview and Notation

This section provides a consolidated overview of the LD-FPG framework's workflow, key mathematical notation used throughout the paper, and supplementary details regarding data processing, model architectures, and training procedures.

## A.1  Notation

We define the following notation used throughout the main text and appendix:

- $k$: Number of nearest neighbors used for graph construction.
- $\tilde{X}^{(t)} \in \mathbb{R}^{N \times 3}$: Raw (unaligned) Molecular Dynamics coordinates for frame $t$.
- $\Theta$: Encoder neural network function; $\Theta^*$ denotes frozen, pre-trained encoder parameters.
- $K$: Order of Chebyshev polynomials in the ChebNet encoder.
- $d_c$: Dimensionality of the conditioner tensor $C$.
- $H, W$: Output dimensions (height and width respectively) for pooling layers, where $d_p = H \cdot W$.
- $N_{\text{layers}}$: Number of layers in a Multi-Layer Perceptron (MLP).
- $\mathcal{L}_{\text{diffusion}}$: Loss function for training the Denoising Diffusion Probabilistic Model (DDPM).
- $N$: Total heavy atoms per structure.
- $V = \{1, ..., N\}$: Atom nodes.
- $G = (V, E)$: Graph representation.
- $\mathcal{I}_{\text{bb}} \subset V$: Backbone atom indices; $N_{\text{bb}} = |\mathcal{I}_{\text{bb}}|$.
- $\mathcal{I}_{\text{sc}} \subset V$: Sidechain atom indices; $N_{\text{sc}} = |\mathcal{I}_{\text{sc}}|$. ($V = \mathcal{I}_{\text{bb}} \cup \mathcal{I}_{\text{sc}}$).
- $z_{\text{res},j} \in \mathbb{R}^{d_p}$: Local pooled context vector summarizing the deformation of residue $R_j$ (this corresponds to $\mathbf{h}_{R_j}^{(b)}$ as used in Appendix D.3).
- $X_{\text{true}} \in \mathbb{R}^{N \times 3}$: Ground truth aligned coordinates. $X_{\text{true}}^{(b,i)}$ for atom $i$, sample $b$.
- $X_{\text{pred}} \in \mathbb{R}^{N \times 3}$: Predicted coordinates.
- $X_{\text{ref}} \in \mathbb{R}^{N \times 3}$: Reference structure coordinates (e.g., first frame).
- $d_z$: Encoder latent embedding dimensionality.
- $Z \in \mathbb{R}^{N \times d_z}$: Latent atom embeddings from encoder: $Z = \text{Encoder}(X_{\text{true}}, E)$.
- $Z_{\text{ref}} \in \mathbb{R}^{N \times d_z}$: Reference structure latent embeddings: $Z_{\text{ref}} = \text{Encoder}(X_{\text{ref}}, E_{\text{ref}})$.
- $C \in \mathbb{R}^{N \times d_c}$: General conditioning tensor (typically $C = Z_{\text{ref}}$). $C_{ex}$ is batch-expanded.
- $d_p$: Pooled context vector dimension.
- $\mathcal{P}$: Generic pooling operator.
- $\mathcal{R} = \{R_1, ..., R_{N_{\text{res}}}\}$: Partition of atoms into residues.
- $f : V \rightarrow \{1, ..., N_{\text{res}}\}$: Atom-to-residue mapping.
- $\phi, \psi, \chi_k$: Dihedral angles; $\alpha \in \mathcal{A}$ is a generic type.
- $M_\alpha(j)$: Mask for valid angle $\alpha$ in residue $j$.
- $P(\alpha_{\text{pred}}), P(\alpha_{\text{true}})$: Empirical angle distributions.
- $D(\cdot \| \cdot)$: Divergence function (KL, JS).
- $w_{\text{base}}, \lambda_{\text{mse}}, \lambda_{\text{div}}$: Loss weights.
- $f_{\text{dih}}$: Probability of applying dihedral-based loss terms during stochastic fine-tuning.
- $\|\mathbf{v}\|^2$: Squared Euclidean norm.
- $T$: Diffusion timesteps.
- $\beta_t, \alpha_t, \bar{\alpha}_t$: Diffusion schedule parameters.

- $\mathbf{h}_0$: Initial pooled latent embedding for diffusion.
- $\mathbf{h}_t$: Noisy latent at step $t$.
- $\epsilon \sim \mathcal{N}(0, I)$: Standard Gaussian noise.
- $\epsilon_\theta(\mathbf{h}_t, t)$: Denoising network predicting noise $\epsilon$.

## A.2 Overall Training and Generation Workflow

The LD-FPG framework follows a multi-phase training and generation procedure, outlined in Algorithm 1.

**Algorithm 1** Overall Training and Generation Workflow (LD-FPG)

---

1: **Input:** Aligned coordinates file (`my_protein.json`), Topology file (`condensed_residues.json`), Reference PDB (`heavy_chain.pdb`).

2: **Parameters:** Encoder config ($d_z, K, \dots$), Decoder config (Pooling Type, $H, W, N_{layers}, \dots$), Optional fine-tuning weights ($\lambda_{mse}, \lambda_{div}$), Diffusion config ($T, \beta_{start}, \beta_{end}, \epsilon_\theta$ type).

  // — **Preprocessing (Details in Appendix B)** —

3: Load aligned coordinates $\{X_{\text{true}}^{(f)}\}_{f=1}^{N_{\text{frames}}}$ from `my_protein.json`.

4: Load topology (atom indices $\mathcal{I}_{\text{bb}}, \mathcal{I}_{\text{sc}}$, residue map $f$, dihedral defs.) from `condensed_residues.json`.

5: Build graph dataset $GraphDataset = \{\text{BuildGraph}(X_{\text{true}}^{(f)}, k = 4)\}_{f=1}^{N_{\text{frames}}}$.

  // — **Phase 1: Encoder Pre-training (Optional) (Details in Appendix ??)** —

6: Initialize Encoder$_\theta$ (ChebNet model).

7: Train Encoder$_\theta$ on $GraphDataset$ using coordinate reconstruction loss $\mathcal{L}_{\text{HNO}}$ (Eq. in Appendix F).

8: Save best encoder parameters $\theta^*$.

  // — **Phase 2: Decoder Training (Details in Appendix D)** —

9: Load pre-trained Encoder$_{\theta*}$ and freeze weights.

10: Generate latent embeddings $Z^{(f)} = \text{Encoder}_{\theta*}(X_{\text{true}}^{(f)})$ for all frames $f$.

11: Create Decoder Input Dataset $D_{dec} = \{(Z^{(f)}, X_{\text{true}}^{(f)}) \mid f = 1..N_{\text{frames}}\}$.

12: Define conditioner $C = Z_{\text{ref}} = \text{Encoder}_{\theta*}(X_{\text{true}}^{(1)})$ (Details in Appendix ??).

13: **Select Decoder Variation:**

14: **if** PoolingType is Blind **then**

15:   Initialize Decoder$_\phi \leftarrow$ BlindPoolingDecoder($\dots$).

16:   Train Decoder$_\phi$ on $D_{dec}$ using coordinate loss $\mathcal{L}_{\text{Dec}} = \mathcal{L}_{\text{coord}}$.

17:   *Optional Fine-tuning:* Continue training with $\mathcal{L}_{\text{Dec}} = w_{\text{base}}\mathcal{L}_{\text{coord}} + \lambda_{mse}\mathcal{L}_{\text{mse\_dih}} + \lambda_{div}\mathcal{L}_{\text{div\_dih}}$ (stochastically, see Appendix F).

18:   Save best decoder parameters $\phi^*$.

19: **else if** PoolingType is Sequential **then**

20:   Initialize BBDecoder$_{\phi_{bb}}$, SCDecoder$_{\phi_{sc}}$($\dots$).

21:   Train BBDecoder$_{\phi_{bb}}$ on $D_{dec}$ using $\mathcal{L}_{\text{BB}} = \mathcal{L}_{\text{coord}}^{\text{bb}}$. Freeze $\phi_{bb}^*$.

22:   Train SCDecoder$_{\phi_{sc}}$ on $D_{dec}$ using $\mathcal{L}_{\text{SC}} = \mathcal{L}_{\text{coord}}^{\text{full}}$ (requires frozen BBDecoder$_{\phi_{bb}^*}$).

23:   Save best parameters $\phi_{bb}^*, \phi_{sc}^*$.

24: **else if** PoolingType is Residue-Based **then**

25:   Initialize Decoder$_\phi \leftarrow$ ResidueBasedDecoder($\dots$).

26:   Train Decoder$_\phi$ on $D_{dec}$ using coordinate loss $\mathcal{L}_{\text{Dec}} = \mathcal{L}_{\text{coord}}$.

27:   Save best decoder parameters $\phi^*$.

28: **end if**

  // — **Phase 3: Latent Diffusion Training (Details in Appendix E)** —

29: Load best Encoder$_{\theta*}$ and Decoder$_{\phi*}$ (or relevant pooling part).

30: Generate pooled latent embeddings $\mathbf{h}_0^{(f)} = \text{Pool}(Z^{(f)})$ for all $f$, using the specific pooling mechanism of the chosen decoder.

31: Create $DiffusionInputDataset = \{\mathbf{h}_0^{(f)}\}$.

32: Initialize $\epsilon_\theta$ (Denoising model).

33: Train $\epsilon_\theta$ on $DiffusionInputDataset$ using $\mathcal{L}_{\text{diffusion}}$ (Eq. 2).

34: Save best diffusion model parameters $\theta_{diff}^*$.

  // — **Output Generation (Sampling)** —

35: Load best models: Encoder$_{\theta*}$, Decoder$_{\phi*}$ (or BBDecoder$_{\phi_{bb}^*}$, SCDecoder$_{\phi_{sc}^*}$), $\epsilon_{\theta_{diff}^*}$.

36: Sample novel pooled latent(s) $\mathbf{h}_0^{\text{gen}}$ using $\epsilon_{\theta_{diff}^*}$ (Algorithm in Appendix E).

37: **Decode generated latent(s):** (Feed $\mathbf{h}_0^{\text{gen}}$ into appropriate decoder stage)

38: $X_{\text{pred}}^{\text{gen}} \leftarrow \text{Decoder}_{\phi*}(\dots, \text{context} = \mathbf{h}_0^{\text{gen}}, C)$

39: **Output:** Generated coordinates $\{X_{\text{pred}}^{\text{gen}}\}$

---

# B   Input Data Processing and Representation

The raw Molecular Dynamics (MD) simulation data (trajectory processing detailed in Appendix G.1) is transformed into a structured format suitable for the machine learning pipeline. This involves generating two key JSON files using custom Python scripts (see Supplementary Code for `extract_residues.py` and `condense_json.py`): one defining the static topology of the protein with a consistent indexing scheme (`condensed_residues.json`), and another containing per-frame coordinate and dihedral angle data.

## B.1   Static Topology and Consistent Indexing File (`condensed_residues.json`)

To provide a consistent structural map for the machine learning models, a static JSON file, typically named `condensed_residues.json`, is generated. This file is crucial as it establishes a definitive and model-centric representation of the protein's topology:

- **Zero-Based Contiguous Atom Indexing:** A new, zero-based, and contiguous indexing scheme $(0, 1, \ldots, N-1)$ is created for all $N$ heavy atoms in the protein system. This re-indexing maps original PDB atom identifiers to a consistent integer range, essential for constructing graph inputs and feature matrices for the neural network.

- **Residue Definitions:** Residues are also re-indexed contiguously (e.g., $0, \ldots, N_{\text{res}} - 1$). For each re-indexed residue, the file stores:
    - The residue type (e.g., 'ALA', 'LYS').
    - Lists of atom indices (using the *new zero-based scheme*) that constitute the backbone atoms of that residue.
    - Lists of atom indices (using the *new zero-based scheme*) that constitute the sidechain heavy atoms of that residue.

- **Dihedral Angle Definitions (Atom Quadruplets):** This is a critical component for calculating dihedral-based losses and analyses. For each residue, the file stores definitions for all applicable standard backbone angles $(\phi, \psi)$ and sidechain angles $(\chi_1, \chi_2, \chi_3, \chi_4, \chi_5)$. Each dihedral angle is defined by an ordered quadruplet of four atom indices. Crucially, these atom indices adhere to the *new, zero-based, contiguous indexing scheme*. For example, a $\phi$ angle for a specific residue would be defined by four specific integer indices from the $0 \ldots N-1$ range. This allows for unambiguous calculation of any dihedral angle directly from a set of $N$ atomic coordinates, whether they are ground truth or model-predicted. The definitions also account for residue types where certain $\chi$ angles are not present (e.g., Alanine has no $\chi$ angles, Glycine has no sidechain).

## B.2   Primary Per-Frame Data File

This second JSON file stores the dynamic information extracted from each frame of the MD trajectory. For every snapshot:

- **Heavy Atom Coordinates:** The 3D Cartesian coordinates of all heavy atoms are recorded after rigid-body alignment to a common reference frame (the first frame of the trajectory, as described in Section 3.1). These coordinates are stored in an order that corresponds to the new zero-based indexing defined in the static topology file.

- **Dihedral Angle Values:** The values for standard backbone dihedral angles $(\phi, \psi)$ and sidechain dihedral angles ($\chi_1$ through $\chi_5$, where applicable for each residue type) are pre-calculated. These calculations initially use atom identifications based on the original PDB residue and atom naming conventions but are stored in a way that can be mapped to the new indexing if needed for direct comparison or analysis.

This primary data file is typically organized on a per-residue basis (using original PDB residue numbering for initial organization if helpful during generation), associating each residue with its constituent atoms' names, original PDB indices, and the time series of their coordinates and calculated dihedral angles.

### B.3   Usage in Models

The two JSON files are used in conjunction:

- The static topology file (`condensed_residues.json`) serves as the definitive reference for all structural metadata used by the model during training and inference. This includes identifying which atoms belong to the backbone versus sidechain (using their new zero-based indices) and, most importantly, providing the specific quadruplets of new zero-based atom indices required to calculate any dihedral angle from a given set of 3D coordinates. This capability is essential for implementing the dihedral angle-based loss terms ($\mathcal{L}_{\mathrm{mse\_dih}}$, $\mathcal{L}_{\mathrm{div\_dih}}$) mentioned in Section 3.5, as these losses operate on dihedral angles computed from the model's predicted coordinates ($X_{\mathrm{pred}}$).

- The 3D coordinates for each heavy atom, required as input features ($X^{(t)}$) for the encoder at each frame $t$, are drawn from the primary per-frame data file. These coordinates must be arranged and ordered according to the *new zero-based indexing scheme* established by the static topology file to ensure consistency with the model's internal graph representation.

This separation of dynamic coordinate data from static, re-indexed topological information allows for efficient data loading and consistent geometric calculations within the LD-FPG framework.

## C   Other validation studies

### C.1   Cross–system coverage on diverse proteins

**Setup.**   To test generality beyond GPCRs, we evaluated LD-FPG on twenty proteins spanning a wide range of lengths and secondary-structure composition (helical, $\beta$-sheet, and mixed $\alpha/\beta$). The systems are: `3dan_A`, `1af7_A`, `3tdt_A`, `1kop_A`, `5jak_A`, `1sdi_A`, `1zt5_A`, `1u7k_D`, `1jyo_E`, `2ppp_A`, `3zcb_B`, `5l87_A`, `3mmy_D`, `1ab1_A`, `few_sec`, `alpha-beta`, `antistatin`, `villin`, `kinase_1ptq`, `7jfl_C`. All reference trajectories are public MD simulations from the ATLAS database. We Kabsch-align each trajectory to its first frame using $C\alpha$ atoms to eliminate rigid-body motion. LD-FPG is trained *per system* on the aligned frames and then used to sample new all-atom conformations that are decoded back to Cartesian coordinates.

**PCA protocol (per system).**   For an interpretable 2D visualization of the conformational manifold, we compute principal components *only from the reference MD*: (i) flatten the aligned $C\alpha$ coordinates for each frame to a vector in $\mathbb{R}^{3N}$, (iii) fit PCA and keep the top two components. We then project (a) a downsample of MD frames (blue points) and (b) an equal number of LD-FPG samples (red points) into that fixed basis. This avoids any leakage from model samples into the PCA axes.

**Observation.**   Across folds and sizes, LD-FPG reproduces the global geometry of the MD manifolds: elongated, anisotropic clouds (e.g., `1af7_A`, `3tdt_A`, `7jfl_C`), multi-lobed structures (`3dan_A`, `1kop_A`), and more compact clusters (`3mmy_D`, `villin`). In some systems the model slightly contracts the most distant extremes (a conservative bias), while in others it marginally broadens dense cores.

### C.2   Validation on GPCR activation:   TM-distance free–energy surface

**Dataset and preprocessing.**   We use a publicly released all-atom, explicit-solvent molecular dynamics (MD) trajectory of the human adenosine $A_1$ receptor embedded in a lipid bilayer [67]. Prior to encoding/decoding, solvent, ions, and lipids are stripped; all protein frames are Kabsch-aligned to a reference using $C\alpha$ atoms from the transmembrane (TM) bundle to remove rigid-body drift. The same alignment and atom selections are applied to both the MD reference and the LD-FPG–generated structures.

**Activation coordinates.**   Following standard GPCR analyses, we quantify activation by two intracellular opening distances: TM3–6 and TM3–7. Each distance is computed between the centers of mass of $C\alpha$ atoms at the cytoplasmic ends of the respective helices (helix assignments by DSSP; residue windows as in D'Amore et al. 67). These coordinates track the outward swing of TM6 and the cavity opening toward the G-protein, i.e., the hallmark GPCR activation motion.

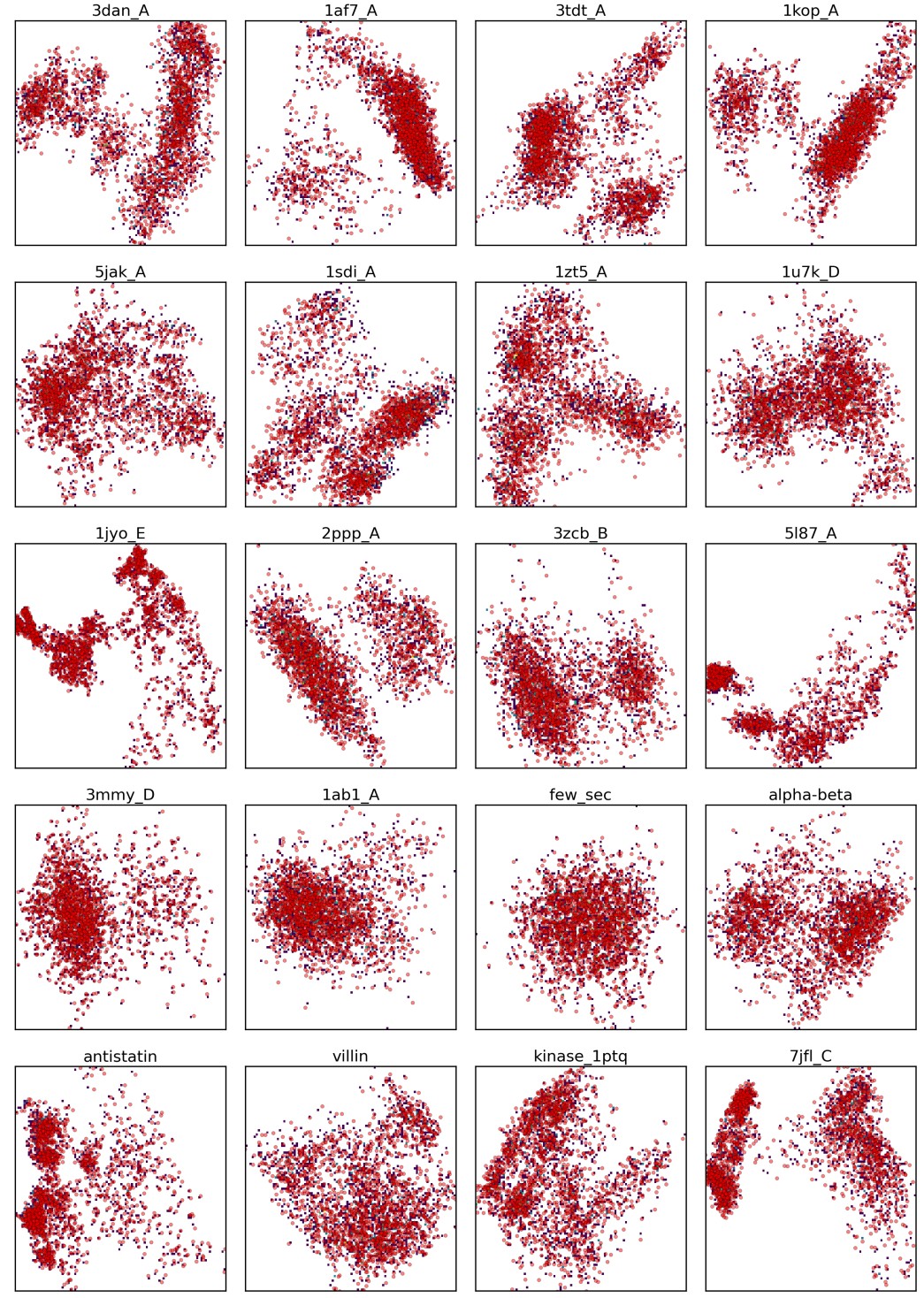

Figure 5: **Diverse-protein PCA coverage.** Top-2 PCA of Cα coordinates computed *from MD only* for each system; MD frames (blue) and LD-FPG samples (red) are projected into the same basis. Panels span proteins of different lengths and secondary-structure content.

**Free-energy estimation.** For both the MD reference and LD-FPG samples, we estimate the joint distribution $P(d_{36}, d_{37})$ by binning the two distances on a regular grid and converting to a potential of mean force (PMF) via $F(d_{36}, d_{37}) = -k_{\mathrm{B}}T \ln P(d_{36}, d_{37}) + C$. One-dimensional profiles are obtained by marginalization, e.g., $F(d_{36}) = -k_{\mathrm{B}}T \ln \int P(d_{36}, d_{37}) \, \mathrm{d}d_{37}$.

**Result.** Figure 6 (left) shows the MD PMF (heatmap) overlaid with LD-FPG contours in the $(d_{36}, d_{37})$ plane; the right panels plot the corresponding one-dimensional PMFs along each coordinate. The reference surface forms a diagonal valley connecting inactive- and active-like regions. LD-FPG reproduces this valley and the position/curvature of the principal minimum, with modest broadening at the valley shoulders. The agreement of the 1D profiles indicates that the model samples along the activation corridor with near-reference occupancy, supporting that LD-FPG captures functionally relevant GPCR motions beyond local geometry.

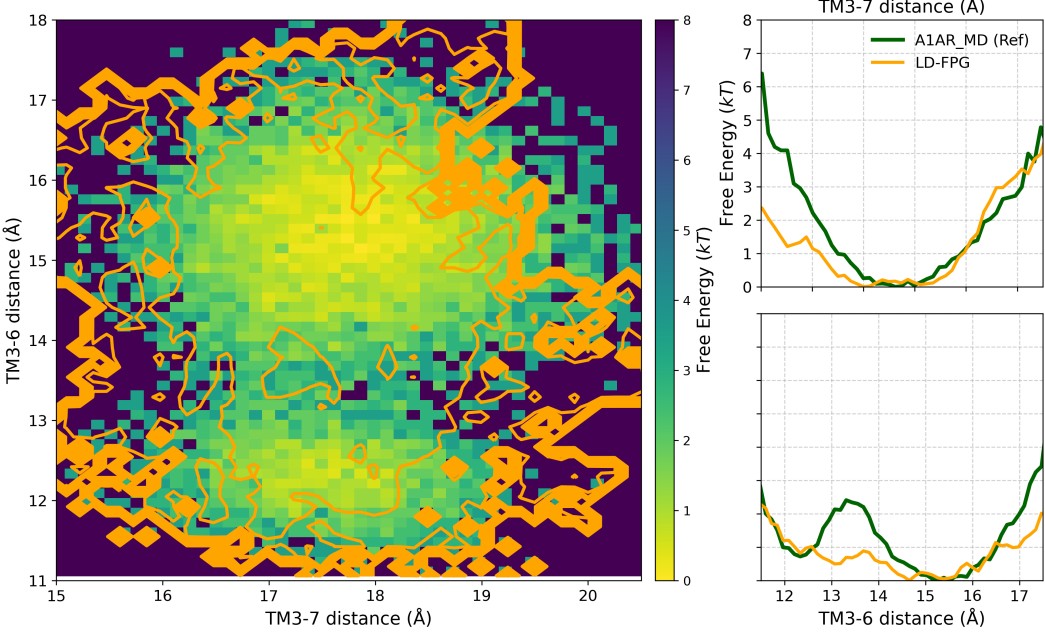

Figure 6: **A1AR activation surface.** *Left:* Two-dimensional PMF over TM3–6 and TM3–7 distances; MD reference as a heatmap, LD-FPG as overlaid contours. *Right:* One-dimensional PMFs along each coordinate (MD in green; LD-FPG in orange). LD-FPG tracks the diagonal activation valley and the principal minimum with minor broadening at the flanks.

# D  Decoder Architectures and Pooling Strategies

This section provides a detailed description of the three primary pooling strategies employed within the decoder architectures: Blind pooling, sequential pooling, and residue-based pooling. For each strategy, we delineate how atom-wise latent embeddings $Z \in \mathbb{R}^{B \times N \times d_z}$ (where $B$ is batch size, $N$ is the number of heavy atoms, and $d_z$ is the latent dimension per atom) and a conditioner $C \in \mathbb{R}^{N \times d_c}$ (typically the latent representation of the reference structure, $Z_{\mathrm{ref}}$) are processed to generate the input for the final coordinate prediction MLP. Specific hyperparameter configurations for representative models are detailed in the Extended Technical Appendix (Section J).

## D.1  Blind Pooling Strategy

The blind pooling strategy aims to capture a global context from all atom embeddings for the entire protein structure. Let $Z^{(b)} \in \mathbb{R}^{N \times d_z}$ be the latent atom embeddings for a single sample $b$ in a batch.

1. **Global Pooling:** The atom embeddings $Z^{(b)}$ are treated as an image-like tensor and processed by a 2D adaptive average pooling layer, $\mathcal{P}_{\mathrm{global}}$ (typically `nn.AdaptiveAvgPool2d` with output size $H \times W$). This operation pools across all $N$ atoms for each sample in the batch:
$$\mathbf{h}_{\mathrm{global}}^{(b)} = \mathcal{P}_{\mathrm{global}}(Z^{(b)}) \in \mathbb{R}^{d_p}$$
where $d_p = H \cdot W$ is the dimension of the pooled global context vector. For a batch, this results in $\mathbf{H}_{\mathrm{global}} \in \mathbb{R}^{B \times d_p}$.

2. **Context Expansion:** This global context vector $\mathbf{h}_{\mathrm{global}}^{(b)}$ is then expanded (tiled) to match the number of atoms $N$, resulting in $\mathbf{H}_{\mathrm{global\_ex}}^{(b)} \in \mathbb{R}^{N \times d_p}$, where each row $i$ (for atom $i$) is identical to $\mathbf{h}_{\mathrm{global}}^{(b)}$. For a batch, this is $\mathbf{H}_{\mathrm{global\_ex}} \in \mathbb{R}^{B \times N \times d_p}$.

3. **Conditioner Expansion:** The conditioner $C \in \mathbb{R}^{N \times d_c}$ is expanded for the batch to $C_{ex} \in \mathbb{R}^{B \times N \times d_c}$.

4. **MLP Input Formulation:** For each atom $i$ in sample $b$, the input to the final MLP, $\mathrm{MLP}_{\mathrm{blind}}$, is formed by concatenating its corresponding expanded global context and its conditioner vector:
$$M_{\mathrm{in}}^{(b,i)} = \mathrm{concat}(\mathbf{h}_{\mathrm{global}}^{(b)}, C^{(b,i)}) \in \mathbb{R}^{d_p + d_c}$$
Note that $\mathbf{h}_{\mathrm{global}}^{(b)}$ is the same for all atoms $i$ within sample $b$.

5. **Coordinate Prediction:** A shared $\mathrm{MLP}_{\mathrm{blind}}$ processes $M_{\mathrm{in}}^{(b,i)}$ for each atom to predict its 3D coordinates:
$$X_{\mathrm{pred}}^{(b,i)} = \mathrm{MLP}_{\mathrm{blind}}(M_{\mathrm{in}}^{(b,i)}) \in \mathbb{R}^3$$
This results in the full predicted structure $X_{\mathrm{pred}} \in \mathbb{R}^{B \times N \times 3}$.

## D.2  Sequential Pooling Strategy

The sequential pooling strategy decodes the protein structure in two stages: first the backbone atoms ($\mathcal{I}_{\mathrm{bb}}$), then the sidechain atoms ($\mathcal{I}_{\mathrm{sc}}$), using information from the preceding stage.

### D.2.1  Backbone Decoder Stage

Let $Z^{(b)} \in \mathbb{R}^{N \times d_z}$ be the full latent atom embeddings and $C^{(b)} \in \mathbb{R}^{N \times d_c}$ be the full conditioner for sample $b$.

1. **Backbone Embedding Selection:** Latent embeddings $Z_{\mathrm{bb}}^{(b)} \in \mathbb{R}^{N_{\mathrm{bb}} \times d_z}$ and conditioner vectors $C_{\mathrm{bb}}^{(b)} \in \mathbb{R}^{N_{\mathrm{bb}} \times d_c}$ corresponding to backbone atoms $\mathcal{I}_{\mathrm{bb}}$ are selected.

2. **Backbone Pooling:** $Z_{\mathrm{bb}}^{(b)}$ is pooled using a 2D adaptive average pooling layer $\mathcal{P}_{\mathrm{bb}}$ (e.g., `BlindPooling2D` from the implementation, with output size $H_{bb} \times W_{bb}$) to obtain a backbone-specific context vector:
$$\mathbf{h}_{\mathrm{bb}}^{(b)} = \mathcal{P}_{\mathrm{bb}}(Z_{\mathrm{bb}}^{(b)}) \in \mathbb{R}^{d_{p,bb}}$$

where $d_{p,bb} = H_{bb} \cdot W_{bb}$. For a batch, this is $\mathbf{H}_{bb} \in \mathbb{R}^{B \times d_{p,bb}}$.

3. **Context and Conditioner Expansion:** $\mathbf{h}_{bb}^{(b)}$ is expanded to $\mathbf{H}_{bb\_ex}^{(b)} \in \mathbb{R}^{N_{bb} \times d_{p,bb}}$. $C_{bb}^{(b)}$ is used directly.

4. **MLP Input for Backbone Atoms:** For each backbone atom $j \in \mathcal{I}_{bb}$ in sample $b$, the input to the backbone MLP, $\text{MLP}_{bb}$, is:

$$M_{\text{in, bb}}^{(b,j)} = \text{concat}(\mathbf{h}_{bb}^{(b)}, C_{bb}^{(b,j)}) \in \mathbb{R}^{d_{p,bb}+d_c}$$

5. **Backbone Coordinate Prediction:** A shared $\text{MLP}_{bb}$ predicts backbone coordinates:

$$X_{\text{pred}_{bb}}^{(b,j)} = \text{MLP}_{bb}(M_{\text{in, bb}}^{(b,j)}) \in \mathbb{R}^3$$

This yields the predicted backbone structure $X_{pred_{bb}} \in \mathbb{R}^{B \times N_{bb} \times 3}$.

### D.2.2 Sidechain Decoder Stage

This stage predicts sidechain atom coordinates $X_{\text{pred}_{sc}}$ using the full latent embeddings $Z^{(b)} \in \mathbb{R}^{N \times d_z}$, the conditioner $C^{(b)} \in \mathbb{R}^{N \times d_c}$, and the predicted backbone coordinates $X_{\text{pred}_{bb}}^{(b)} \in \mathbb{R}^{B \times N_{bb} \times 3}$ from the Backbone Decoder Stage.

1. **Sidechain Embedding Selection:** Latent embeddings $Z_{sc}^{(b)} \in \mathbb{R}^{N_{sc} \times d_z}$ corresponding to sidechain atoms $\mathcal{I}_{sc}$ are selected from the full latent embeddings $Z^{(b)}$.

2. **Sidechain Pooling:** The selected sidechain embeddings $Z_{sc}^{(b)}$ are pooled using a 2D adaptive average pooling layer $\mathcal{P}_{sc}$ (e.g., `BlindPooling2D` with output size $H_{sc} \times W_{sc}$) to obtain a sidechain-specific context vector for each sample in the batch:

$$\mathbf{h}_{sc}^{(b)} = \mathcal{P}_{sc}(Z_{sc}^{(b)}) \in \mathbb{R}^{d_{p,sc}}$$

where $d_{p,sc} = H_{sc} \cdot W_{sc}$. For a batch, this results in $\mathbf{H}_{sc} \in \mathbb{R}^{B \times d_{p,sc}}$.

3. **Feature Construction for Sidechain MLP Input:** The input to the sidechain MLP, $\text{MLP}_{sc}$, is a global vector $M_{\text{in, sc}}^{(b)}$ constructed per sample $b$. The construction varies based on the `arch_type`:

   - Let $X_{\text{pred}_{bb\_flat}}^{(b)} \in \mathbb{R}^{N_{bb} \cdot 3}$ be the flattened predicted backbone coordinates for sample $b$.
   - Let $C_{sc}^{(b)} \in \mathbb{R}^{N_{sc} \times d_c}$ be the sidechain portion of the reference conditioner. If $d_c = d_z$, this is $Z_{\text{ref, sc}}^{(b)}$. This is flattened to $C_{sc\_flat}^{(b)} \in \mathbb{R}^{N_{sc} \cdot d_c}$.

   **Arch-Type 0:** The input consists of the flattened predicted backbone coordinates and the pooled sidechain context from the current frame's embeddings.

$$M_{\text{in, sc}}^{(b)}[\text{Arch } 0] = \text{concat}(X_{\text{pred}_{bb\_flat}}^{(b)}, \mathbf{h}_{sc}^{(b)})$$

   The dimension of $M_{\text{in, sc}}^{(b)}[\text{Arch } 0]$ is $(N_{bb} \cdot 3) + d_{p,sc}$.
   **Arch-Type 1:** This architecture adds a reduced representation of the sidechain portion of the reference conditioner.

   - The flattened sidechain conditioner $C_{sc\_flat}^{(b)}$ is passed through a linear reduction layer:

$$C_{sc\_reduced}^{(b)} = \text{Linear}_{sc\_reduce}(C_{sc\_flat}^{(b)}) \in \mathbb{R}^{d'_{c,sc}}$$

   (e.g., $d'_{c,sc} = 128$ in the implementation).
   The MLP input is then:

$$M_{\text{in, sc}}^{(b)}[\text{Arch } 1] = \text{concat}(X_{\text{pred}_{bb\_flat}}^{(b)}, \mathbf{h}_{sc}^{(b)}, C_{sc\_reduced}^{(b)})$$

   The dimension of $M_{\text{in, sc}}^{(b)}[\text{Arch } 1]$ is $(N_{bb} \cdot 3) + d_{p,sc} + d'_{c,sc}$.
   **Arch-Type 2:** This architecture uses a reduced representation of both the predicted backbone coordinates and the sidechain portion of the reference conditioner.

- The flattened predicted backbone coordinates $X_{\text{pred}_{\text{bb\_flat}}}^{(b)}$ are passed through a linear reduction layer:

$$X_{\text{pred}_{\text{bb\_reduced}}}^{(b)} = \text{Linear}_{\text{bb\_reduce}}(X_{\text{pred}_{\text{bb\_flat}}}^{(b)}) \in \mathbb{R}^{d'_{\text{bb}}}$$

(e.g., $d'_{\text{bb}} = 128$ in the implementation).

- The sidechain conditioner is reduced as in Arch-Type 1 to $C_{\text{sc\_reduced}}^{(b)} \in \mathbb{R}^{d'_{c,sc}}$.

The MLP input is then:

$$M_{\text{in, sc}}^{(b)}[\text{Arch 2}] = \text{concat}(X_{\text{pred}_{\text{bb\_reduced}}}^{(b)}, \mathbf{h}_{\text{sc}}^{(b)}, C_{\text{sc\_reduced}}^{(b)})$$

The dimension of $M_{\text{in, sc}}^{(b)}[\text{Arch 2}]$ is $d'_{\text{bb}} + d_{p,sc} + d'_{c,sc}$.

4. **Sidechain Coordinate Prediction:** The sidechain MLP, $\text{MLP}_{\text{sc}}$, processes the constructed input vector $M_{\text{in, sc}}^{(b)}$ (corresponding to the chosen `arch_type`) to predict all sidechain coordinates for sample $b$ simultaneously:

$$X_{\text{pred}_{\text{sc\_flat}}}^{(b)} = \text{MLP}_{\text{sc}}(M_{\text{in, sc}}^{(b)}) \in \mathbb{R}^{N_{\text{sc}} \cdot 3}$$

This flattened output is then reshaped to $X_{\text{pred}_{\text{sc}}}^{(b)} \in \mathbb{R}^{N_{\text{sc}} \times 3}$.

5. **Full Structure Assembly:** The final predicted structure $X_{\text{pred}}^{(b)} \in \mathbb{R}^{N \times 3}$ for sample $b$ is assembled by combining the predicted backbone coordinates $X_{\text{pred}_{\text{bb}}}^{(b)}$ and the predicted sidechain coordinates $X_{\text{pred}_{\text{sc}}}^{(b)}$.

## D.3 Residue-based Pooling Strategy

The Residue-based Pooling strategy generates a context vector specific to each residue and uses this local context for predicting the coordinates of atoms within that residue. Let $Z^{(b)} \in \mathbb{R}^{N \times d_z}$ be the latent atom embeddings for sample $b$. Let $V_j$ be the set of atom indices belonging to residue $R_j$, and $f : V \to \{1, ..., N_{\text{res}}\}$ be the mapping from a global atom index to its residue index.

1. **Per-Residue Pooling:** For each residue $R_j$ in sample $b$:
   - Select atom embeddings for residue $R_j$: $Z_{R_j}^{(b)} \in \mathbb{R}^{|V_j| \times d_z}$.
   - Pool these embeddings using a 2D adaptive average pooling layer $\mathcal{P}_{\text{res}}$ (e.g., `nn.AdaptiveAvgPool2d` with output $H \times W$):

$$\mathbf{h}_{R_j}^{(b)} = \mathcal{P}_{\text{res}}(Z_{R_j}^{(b)}) \in \mathbb{R}^{d_p}$$

   where $d_p = H \cdot W$.

   This results in a set of $N_{\text{res}}$ pooled vectors for sample $b$, which can be represented as $\mathbf{H}_{\text{res}}^{(b)} \in \mathbb{R}^{N_{\text{res}} \times d_p}$. For a batch, this is $\mathbf{H}_{\text{res}} \in \mathbb{R}^{B \times N_{\text{res}} \times d_p}$.

2. **Atom-Specific Context Assembly:** For each atom $i$ in sample $b$, its specific context vector is the pooled vector of its parent residue $R_{f(i)}$:

$$\mathbf{h}_{\text{atom\_context}}^{(b,i)} = \mathbf{h}_{R_{f(i)}}^{(b)} \in \mathbb{R}^{d_p}$$

   This can be gathered for all atoms to form $\mathbf{H}_{\text{atom\_context}}^{(b)} \in \mathbb{R}^{N \times d_p}$.

3. **Conditioner Expansion:** The conditioner $C \in \mathbb{R}^{N \times d_c}$ is expanded for the batch to $C_{ex} \in \mathbb{R}^{B \times N \times d_c}$.

4. **MLP Input Formulation:** For each atom $i$ in sample $b$, the input to the final MLP, $\text{MLP}_{\text{res}}$, is formed by concatenating its residue's pooled context and its specific conditioner vector:

$$M_{\text{in, res}}^{(b,i)} = \text{concat}(\mathbf{h}_{\text{atom\_context}}^{(b,i)}, C^{(b,i)}) \in \mathbb{R}^{d_p + d_c}$$

5. **Coordinate Prediction:** A shared $\text{MLP}_{\text{res}}$ processes $M_{\text{in, res}}^{(b,i)}$ for each atom to predict its 3D coordinates:

$$X_{\text{pred}}^{(b,i)} = \text{MLP}_{\text{res}}(M_{\text{in, res}}^{(b,i)}) \in \mathbb{R}^3$$

   This results in the full predicted structure $X_{\text{pred}} \in \mathbb{R}^{B \times N \times 3}$.

This strategy allows the model to learn representations that are localized at the residue level, potentially capturing residue-specific conformational preferences more directly.

# E  Latent Diffusion Model Details

This section details the Denoising Diffusion Probabilistic Model (DDPM) [47] utilized in our framework. The DDPM operates on the pooled latent embeddings $\mathbf{h}_0 \in \mathbb{R}^{d_p}$ (where $d_p$ is the dimension of the pooled latent space, dependent on the pooling strategy) obtained from the encoder and pooling stages. Specific architectures and hyperparameters for the denoising network $\epsilon_\theta$ are discussed in the Extended Technical Appendix (Section J).

## E.1  DDPM Formulation

The DDPM consists of a predefined forward noising process and a learned reverse denoising process.

**Forward Process (Noising):** The forward process gradually adds Gaussian noise to an initial latent embedding $\mathbf{h}_0$ over $T$ discrete timesteps. At each timestep $t$, the transition is defined by:

$$q(\mathbf{h}_t|\mathbf{h}_{t-1}) = \mathcal{N}(\mathbf{h}_t; \sqrt{1-\beta_t}\mathbf{h}_{t-1}, \beta_t\mathbf{I})$$

where $\{\beta_t\}_{t=1}^{T}$ is a predefined variance schedule (e.g., linear, cosine) that controls the noise level at each step. A useful property of this process is that we can sample $\mathbf{h}_t$ at any arbitrary timestep $t$ directly from $\mathbf{h}_0$:

$$q(\mathbf{h}_t|\mathbf{h}_0) = \mathcal{N}(\mathbf{h}_t; \sqrt{\bar{\alpha}_t}\mathbf{h}_0, (1-\bar{\alpha}_t)\mathbf{I})$$

where $\alpha_t = 1 - \beta_t$ and $\bar{\alpha}_t = \prod_{s=1}^{t} \alpha_s$. As $t \to T$, if the schedule is chosen appropriately, $\mathbf{h}_T$ approaches an isotropic Gaussian distribution $\mathcal{N}(0, \mathbf{I})$.

**Reverse Process (Denoising):** The reverse process aims to learn the transition $q(\mathbf{h}_{t-1}|\mathbf{h}_t)$, which is intractable directly. Instead, a neural network, $\epsilon_\theta(\mathbf{h}_t, t)$, is trained to predict the noise component $\epsilon$ that was added to $\mathbf{h}_0$ to produce $\mathbf{h}_t = \sqrt{\bar{\alpha}_t}\mathbf{h}_0 + \sqrt{1-\bar{\alpha}_t}\epsilon$, where $\epsilon \sim \mathcal{N}(0, \mathbf{I})$. The network is optimized by minimizing the simplified DDPM loss function (as shown in Eq. 2 in the main text):

$$\mathcal{L}_{\text{diffusion}}(\theta) = \mathbb{E}_{t,\mathbf{h}_0,\epsilon}\left[\|\epsilon - \epsilon_\theta(\sqrt{\bar{\alpha}_t}\mathbf{h}_0 + \sqrt{1-\bar{\alpha}_t}\epsilon, t)\|^2\right]$$

where $t$ is sampled uniformly from $\{1, ..., T\}$.

## E.2  Sampling New Latent Embeddings

Once the denoising network $\epsilon_\theta$ is trained, new latent embeddings $\mathbf{h}_0^{\text{gen}}$ can be generated by starting with a sample from the prior distribution, $\mathbf{h}_T \sim \mathcal{N}(0, \mathbf{I})$, and iteratively applying the reverse denoising step:

$$\mathbf{h}_{t-1} = \frac{1}{\sqrt{\alpha_t}}\left(\mathbf{h}_t - \frac{\beta_t}{\sqrt{1-\bar{\alpha}_t}}\epsilon_\theta(\mathbf{h}_t, t)\right) + \sigma_t\mathbf{z}$$

where $\mathbf{z} \sim \mathcal{N}(0, \mathbf{I})$ for $t > 1$, and $\mathbf{z} = 0$ for $t = 1$. The variance $\sigma_t^2$ is typically set to $\beta_t$ or $\tilde{\beta}_t = \frac{1-\bar{\alpha}_{t-1}}{1-\bar{\alpha}_t}\beta_t$. The full sampling procedure is outlined in Algorithm 2.

---

**Algorithm 2** Reverse Diffusion Sampling for Latent Embeddings

---

1: **Input:** Trained denoising model $\epsilon_\theta$, number of generation samples $B_{\text{gen}}$, dimension of pooled latent $d_p$, diffusion timesteps $T$, schedule parameters (e.g., $\beta_1, ..., \beta_T$).
2: Calculate $\alpha_t = 1 - \beta_t$ and $\bar{\alpha}_t = \prod_{s=1}^{t} \alpha_s$ for all $t$.
3: Set $\sigma_t^2 = \beta_t$ (or alternative like $\tilde{\beta}_t$).
4: Sample initial noise $\mathbf{h}_T \sim \mathcal{N}(0, \mathbf{I})$ of shape $(B_{\text{gen}}, d_p)$.
5: **for** $t = T, ..., 1$ **do**
6:    Sample $\mathbf{z} \sim \mathcal{N}(0, \mathbf{I})$ of shape $(B_{\text{gen}}, d_p)$ if $t > 1$, else $\mathbf{z} = \mathbf{0}$.
7:    Predict noise: $\epsilon_{\text{pred}} \leftarrow \epsilon_\theta(\mathbf{h}_t, t)$
8:    Calculate conditional mean: $\mu_\theta(\mathbf{h}_t, t) = \frac{1}{\sqrt{\alpha_t}}\left(\mathbf{h}_t - \frac{\beta_t}{\sqrt{1-\bar{\alpha}_t}}\epsilon_{\text{pred}}\right)$
9:    Update latent sample: $\mathbf{h}_{t-1} \leftarrow \mu_\theta(\mathbf{h}_t, t) + \sigma_t\mathbf{z}$
10: **end for**
11: **Output:** Generated latent embeddings $\mathbf{h}_0^{\text{gen}} = \mathbf{h}_0$

---

# F    Loss Function Formulations

This section provides the detailed mathematical formulations for the loss functions used in the LD-FPG framework. The notation used is consistent with Appendix A.1.

## F.1    Encoder Pre-training Loss

The pre-training phase for the ChebNet encoder (Encoder$_\theta$) can be performed using a direct coordinate reconstruction head (MLP$_{\text{HNO}}$). The loss function $\mathcal{L}_{\text{HNO}}$ is the Mean Squared Error (MSE) between the predicted coordinates and the ground truth coordinates:

$$\mathcal{L}_{\text{HNO}} = \mathbb{E}_{(X_{\text{true}}, Z) \sim \mathcal{D}} \left[ \|\text{MLP}_{\text{HNO}}(Z) - X_{\text{true}}\|_F^2 \right] \tag{4}$$

where $Z = \text{Encoder}_\theta(X_{\text{true}}, E)$, $\mathcal{D}$ is the training dataset, and $\|\cdot\|_F^2$ denotes the squared Frobenius norm (sum of squared element-wise differences). For a single sample with $N$ atoms, this is $\frac{1}{N} \sum_{i=1}^N \|\text{MLP}_{\text{HNO}}(Z_i) - (X_{\text{true}})_i\|^2$.

## F.2    Decoder Loss Functions

### F.2.1    Coordinate Mean Squared Error ($\mathcal{L}_{\text{coord}}$)

This is the fundamental loss for all decoder architectures, measuring the MSE between predicted coordinates $X_{\text{pred}}$ and ground truth coordinates $X_{\text{true}}$:

$$\mathcal{L}_{\text{coord}} = \mathbb{E}_{(X_{\text{true}}, X_{\text{pred}}) \sim \mathcal{D}_{\text{dec}}} \left[ \|X_{\text{pred}} - X_{\text{true}}\|_F^2 \right] \tag{5}$$

For a single sample, this is $\frac{1}{N} \sum_{i=1}^N \|(X_{\text{pred}})_i - (X_{\text{true}})_i\|^2$.

### F.2.2    Dihedral Angle Mean Squared Error ($\mathcal{L}_{\text{mse\_dih}}$)

This loss penalizes the squared difference between predicted dihedral angles ($\alpha_{\text{pred}}$) and true dihedral angles ($\alpha_{\text{true}}$). It is used *only* for fine-tuning the Blind Pooling decoder.

$$\mathcal{L}_{\text{mse\_dih}} = \sum_{\alpha \in \mathcal{A}} \mathbb{E}_{(b,j)|M_\alpha(j)} \left[ (\alpha_{\text{pred}}^{(b,j)} - \alpha_{\text{true}}^{(b,j)})^2 \right] \tag{6}$$

where $\mathcal{A}$ is the set of all considered dihedral angle types (e.g., $\phi, \psi, \chi_1, \ldots, \chi_5$), $M_\alpha(j)$ is a mask indicating if angle type $\alpha$ is valid for residue $j$ in sample $b$. The expectation is over valid angles in the batch.

### F.2.3    Dihedral Angle Distribution Divergence ($\mathcal{L}_{\text{div\_dih}}$)

This loss encourages the empirical distribution of predicted dihedral angles ($P(\alpha_{\text{pred}})$) to match that of the true angles ($P(\alpha_{\text{true}})$). It is used *only* for fine-tuning the blind pooling decoder.

$$\mathcal{L}_{\text{div\_dih}} = \sum_{\alpha \in \mathcal{A}} D_{\text{KL}}(P(\alpha_{\text{pred}})\|P(\alpha_{\text{true}})) \tag{7}$$

where $D_{\text{KL}}$ is the Kullback-Leibler divergence (or optionally Jensen-Shannon divergence, JSD, or Wasserstein Distance, WD, as specified in hyperparameters). The distributions $P(\cdot)$ are typically estimated from histograms of angles within a batch or across a larger set of samples.

### F.2.4    Combined Decoder Loss ($\mathcal{L}_{\text{Dec}}$)

The definition of $\mathcal{L}_{\text{Dec}}$ depends on the pooling strategy:

**Blind Pooling Decoder:**

- **Initial Training:** The decoder is trained solely on coordinate MSE:

$$\mathcal{L}_{\text{Dec}}^{\text{Blind, initial}} = \mathcal{L}_{\text{coord}}$$

- **Fine-tuning (Optional):** The loss becomes a weighted sum, where dihedral terms are applied stochastically (e.g., to $10\%$ of mini-batches, controlled by $f_{\text{dih}}$):

$$\mathcal{L}_{\text{Dec}}^{\text{Blind, fine-tune}} = w_{\text{base}} \cdot \mathcal{L}_{\text{coord}} + S \cdot (\lambda_{\text{mse}} \cdot \mathcal{L}_{\text{mse\_dih}} + \lambda_{\text{div}} \cdot \mathcal{L}_{\text{div\_dih}})$$

where $S = 1$ with probability $f_{\text{dih}}$ and $S = 0$ otherwise. $w_{\text{base}}$, $\lambda_{\text{mse}}$, and $\lambda_{\text{div}}$ are scalar weights.

**Residue-based Pooling Decoder:** This decoder is trained using only the coordinate MSE loss:

$$\mathcal{L}_{\text{Dec}}^{\text{Residue}} = \mathcal{L}_{\text{coord}}$$

### F.2.5 Sequential Pooling Decoder Losses

The sequential pooling strategy uses two separate MSE-based losses:

- **Backbone Decoder Loss ($\mathcal{L}_{\textbf{BB}}$):** This is the coordinate MSE loss applied specifically to the predicted backbone atom coordinates $X_{\text{predbb}}$ against the true backbone coordinates $X_{\text{truebb}}$:

$$\mathcal{L}_{\text{BB}} = \mathbb{E}_{(X_{\text{truebb}}, X_{\text{predbb}}) \sim \mathcal{D}_{\text{dec}}} \left[ \| X_{\text{predbb}} - X_{\text{truebb}} \|_F^2 \right]$$

This corresponds to applying $\mathcal{L}_{\text{coord}}$ only to atoms $i \in \mathcal{I}_{\text{bb}}$.

- **Sidechain Decoder (Full Structure) Loss ($\mathcal{L}_{\textbf{SC}}$):** After the Sidechain Decoder predicts sidechain coordinates and assembles the full structure $X_{\text{pred}}$, this loss is the coordinate MSE for the entire protein structure:

$$\mathcal{L}_{\text{SC}} = \mathbb{E}_{(X_{\text{true}}, X_{\text{pred}}) \sim \mathcal{D}_{\text{dec}}} \left[ \| X_{\text{pred}} - X_{\text{true}} \|_F^2 \right]$$

This is equivalent to $\mathcal{L}_{\text{coord}}$ applied to the output of the complete two-stage sequential decoder.

Neither $\mathcal{L}_{\text{BB}}$ nor $\mathcal{L}_{\text{SC}}$ include dihedral angle terms in their standard formulation for the sequential pooling decoder.

## G  Experimental Setup Details

This appendix provides further details on the dataset, evaluation metrics, and implementation aspects of the experimental setup.

### G.1  Dataset and MD Simulation Protocol

The conformational dataset for the human Dopamine D2 receptor (D2R) was generated from all-atom Molecular Dynamics (MD) simulations. **System Preparation:** Simulations were initiated from the cryo-EM structure of the D2R in complex with the inverse agonist risperidone (PDB ID: 6CM4 [68]). The risperidone ligand was removed, and the third intracellular loop (ICL3), which is typically flexible or unresolved, was remodeled using RosettaRemodel [69] to represent an apo-like state. The final remodeled D2R structure consisted of 273 residues, comprising 2191 heavy atoms after selection for the simulation system; hydrogen atoms were not explicitly included as input features to our generative model, which focuses on heavy-atom representations. The D2R protein was then embedded in a 1-palmitoyl-2-oleoyl-sn-glycero-3-phosphocholine (POPC) lipid bilayer using the CHARMM-GUI *Membrane Builder* [70]. The system was solvated with TIP3P water [71] and neutralized with 0.15 M NaCl ions. **Simulation Parameters:** The CHARMM36m force field [72] was employed for all protein, lipid, and ion parameters. Simulations were performed using GRO-MACS 2024.2 [73]. The system underwent energy minimization followed by a multi-step equilibration protocol involving NVT and NPT ensembles with position restraints on the protein heavy atoms, which were gradually released. Production simulations were run under the NPT ensemble at 303.15 K (using the v-rescale thermostat [74]) and 1.0 bar (using the C-rescale barostat [75] with semi-isotropic coupling). A 2 fs timestep was used, with LINCS algorithm [76] constraining bonds involving hydrogen atoms. Electrostatic interactions were calculated using the Particle Mesh Ewald (PME) method [77]. **Trajectory Processing:** Ten independent production replicas, each 2 $\mu$s in length, were generated. One replica exhibiting representative dynamics was selected for this study.

The initial $\approx$ 776 ns of this replica were discarded as extended equilibration, yielding a final analysis trajectory of $\approx$ 1.224 $\mu$s. From this, 12,241 frames were sampled at a regular interval of 100 ps. All protein heavy-atom coordinates in these frames were then rigidly aligned to the heavy atoms of the first frame using the Kabsch algorithm [56] to remove global translation and rotation. **Data Splitting and Preprocessing:** The aligned coordinate dataset was split into a training set (90%, 11,017 frames) and a test set (10%, 1,224 frames) chronologically. Static topological information, including lists of backbone and sidechain atom indices based on a consistent re-indexing scheme, and the definitions of atom quadruplets for standard dihedral angles ($\phi, \psi, \chi_1 - \chi_5$), was extracted once from the reference PDB structure. This information was stored in JSON format as detailed in Appendix B.

## G.2 Architectural rationale: spectral GNNs under alignment

We pair a *non-equivariant* spectral GNN (ChebNet) with an *align–then–learn* protocol: each MD frame is Kabsch-aligned to $X_{\text{ref}}$ (Section 3.1; [56]), so the encoder learns *internal* deformations (helices, loops, side chains) rather than rigid-body drift, and the decoder predicts deformations *relative* to $Z_{\text{ref}}$ (Section 3.3), which empirically preserves stereochemistry. ChebNet's Chebyshev–Laplacian filters provide $K$-hop aggregation in one layer and scale as $\mathcal{O}(K|E|)$ with $|E| \approx kN$ on $k$-NN all-atom graphs ([57]); this mitigates over-smoothing/over-squashing seen in deep local message passing ([78–80]) and avoids the $\mathcal{O}(N^2)$ compute/memory of SE(3)-equivariant attention stacks unless heavily sparsified or coarse-grained ([81–84]). Empirically, the encoder is *not* the bottleneck: our ChebNet autoencoder is near-lossless (backbone MSE $\sim 8 \times 10^{-4}$), whereas end-to-end generative error after pooling and diffusion is $\sim 1 \times 10^{-1}$ (Appendix H); thus dominant errors come from deliberate pooling compression and diffusion, not the encoder.

## G.3 Evaluation Metrics

Model performance was assessed using the following metrics:

### G.3.1 Coordinate Accuracy

- **Mean Squared Error (MSE$_{\text{bb}}$, MSE$_{\text{sc}}$):** Calculated as the average squared Euclidean distance between predicted and ground truth coordinates for corresponding atoms. MSE$_{\text{bb}}$ considers C$\alpha$ atoms (or all backbone heavy atoms N, CA, C, O, as specified in implementation) and MSE$_{\text{sc}}$ considers all sidechain heavy atoms. The MSE for a set of $N_k$ atoms (either backbone or sidechain) is: $\text{MSE} = \frac{1}{N_k} \sum_{i=1}^{N_k} \|\mathbf{x}_{\text{pred},i} - \mathbf{x}_{\text{true},i}\|^2$.

- **Local Distance Difference Test (lDDT)** [85]: lDDT evaluates the preservation of local interatomic distances. For each atom, it considers all other atoms within a defined cutoff radius (e.g., 15 Å) in the reference (true) structure. It then calculates the fraction of these interatomic distances that are preserved in the predicted structure within certain tolerance thresholds (e.g., 0.5, 1, 2, and 4 Å). The final lDDT score is an average over these fractions and all residues/atoms. We report lDDT$_{\text{All}}$ (all heavy atoms on backbone and sidechain) and lDDT$_{\text{BB}}$ (backbone heavy atoms). Scores range from 0 to 1, with 1 indicating perfect preservation.

- **Template Modeling score (TM-score)** [86]: TM-score measures the global structural similarity between a predicted model and a reference structure. It is designed to be more sensitive to correct global topology and less sensitive to local errors than RMSD, and its value is normalized to be between 0 and 1, where 1 indicates a perfect match. A TM-score 0.5 generally indicates that the two proteins share a similar fold.

### G.3.2 Distributional Accuracy

- **Summed Kullback–Leibler Divergence ($\sum$KL) and Jensen–Shannon Divergence ($\sum$JSD):** These metrics quantify the similarity between the 1D distributions of predicted and ground truth dihedral angles ($\phi, \psi,$ and $\chi_1$ through $\chi_5$). For each angle type, empirical probability distributions are estimated from histograms (e.g., using 36 bins over the range $[-\pi, \pi]$). The KL or JS divergence is calculated for each angle type, and the reported $\sum$KL or $\sum$JSD is the sum of these divergences over all seven angle types. Lower values indicate higher similarity between the distributions.

### G.3.3 Physical Plausibility

- **Average Steric Clash Counts:** A steric clash is defined as a pair of non-bonded heavy atoms being closer than a specified distance cutoff. For our analysis, we used a cutoff of 2.1 Å. The clash score is the average number of such clashing pairs per generated structure. This metric was computed using BioPython [87] and SciPy's [88] cKDTree for efficient neighbor searching.

### G.3.4 Conformational Landscape Sampling

- **A100 Activation Index Value:** The A100 value is a collective variable developed by Ibrahim et al. [58] to quantify the activation state of Class A G-protein-coupled receptors (GPCRs). It is a linear combination of five specific interhelical $C\alpha$-$C\alpha$ distances that are known to change upon GPCR activation. The formula is given by:

$$A^{100} = -14.43 \times r(V^{1.53}\text{–}L^{7.55}) - 7.62 \times r(D^{2.50}\text{–}T^{3.37})$$
$$+ 9.11 \times r(N^{3.42}\text{–}I^{4.42}) - 6.32 \times r(W^{5.66}\text{–}A^{6.34})$$
$$- 5.22 \times r(L^{6.58}\text{–}Y^{7.35}) + 278.88$$

where $r(X^{BW_1}\text{–}Y^{BW_2})$ denotes the distance in Angstroms between the $C\alpha$ atoms of residue X at Ballesteros-Weinstein (BW) position $BW_1$ and residue Y at BW position $BW_2$. Higher A100 values typically correspond to more active-like states. We use this metric to compare the conformational landscape explored by generated ensembles against the reference MD simulation.

- **Principal Component Analysis (PCA) of Latent Embeddings:** PCA is applied to the set of pooled latent embeddings ($h_0$) generated by the diffusion model and those derived from the MD dataset. Projecting these high-dimensional embeddings onto the first few principal components allows for a 2D visualization, which helps assess whether the generative model captures the diversity and main modes of variation present in the training data's latent space.

### G.3.5 Training Loss Reporting

- **Auxiliary Dihedral MSE Term ($\sum \mathcal{L}_{\text{dih}}$ MSE):** When dihedral losses are active during the fine-tuning of the blind pooling decoder (see Appendix F for $\mathcal{L}_{\text{mse\_dih}}$), we report the sum of the training MSE losses for all considered dihedral angles. This provides insight into how well the model fits these auxiliary geometric targets.

### G.4 Training resources

The input Molecular Dynamics (MD) data was generated using GROMACS 2024.2 [73] on NVIDIA L40S GPUs. With a simulation rate of approximately 250 ns/day, the 2 micro seconds trajectory used in this study required about 8 days of computation. Following data generation, all machine learning models were implemented in PyTorch [59] and trained using the Adam optimizer [60] on NVIDIA L40S and H100 GPUs. The ChebNet autoencoder training required approximately 5,000 epochs at roughly 8 seconds per epoch. Decoder training varied by pooling strategy: blind pooling decoders trained for 3,000-6,000 epochs at approximately 3 seconds per epoch, while residue pooling decoders took about 24 seconds per epoch for a similar number of epochs. Diffusion model training was the most computationally intensive: sequential and blind pooling models trained for around 10,000 epochs, and residue pooling models for up to 150,000 epochs, with each epoch averaging approximately 1 second on L40S GPUs. The machine learning aspects presented in this paper required approximately 15,000 GPU hours, with total experimentation including preliminary setups amounting to roughly 30,000 GPU hours. This substantial GPU usage translates to significant energy consumption and associated carbon emissions; while precise quantification depends on factors like data center Power Usage Effectiveness (PUE) and local energy grid carbon intensity (and was not performed for this study), we acknowledge this environmental cost. Further details on computational resources, including specific MD simulation parameters and machine learning training runtimes for hyperparameter sweeps, are provided in the Extended Technical Appendix (Appendix J). The overall three-phase training and generation workflow (encoder pre-training, decoder training, and diffusion model training) is detailed in Appendix A (Algorithm 1).

## H   Encoder Performance Summary

The ChebNet encoder's reconstruction head performance provides an upper bound on achievable fidelity. Table 4 summarizes key metrics for different latent dimensions ($d_z$). The encoder accurately captures structural features and dihedral angle distributions with minimal deviation from the ground truth MD data.

Table 4: Encoder Reconstruction Head Performance.

| Encoder Config | lDDT$_\text{All}$ ↑ | lDDT$_\text{BB}$ ↑ | TM$_\text{All}$ ↑ | $\sum$JSD$_\text{bb}$ ↓ | $\sum$JSD$_\text{sc}$ ↓ | MSE$_\text{bb}$ ↓ | MSE$_\text{sc}$ ↓ | $\sum\mathcal{L}_\text{dih}$ MSE ↓ |
|---|---|---|---|---|---|---|---|---|
| Encoder ($d_z = 4$) | 0.692 | 0.777 | 0.959 | 0.00005 | 0.0002 | 0.0040 | 0.00715 | 0.00066 |
| Encoder ($d_z = 8$) | 0.697 | 0.780 | 0.959 | 0.00069 | 0.0009 | 0.0021 | 0.00375 | 0.00188 |
| Encoder ($d_z = 16$) | 0.696 | 0.782 | 0.959 | 0.00009 | 0.00016 | 0.0008 | 0.00160 | 0.00058 |
| GT (MD) Ref | 0.698 | 0.779 | 0.959 | - | - | - | - | - |

## I   Residue Pooling Inference Details

For the residue pooling model visualizations (Fig. 2c and Fig. 4c in the main text), samples were aggregated from 10 distinct model checkpoints saved at different stages of the diffusion model's training. This multi-epoch sampling strategy, inspired by moving average techniques [89], was employed to provide a more stable and representative visualization of the learned conformational space, averaging out potential epoch-specific biases.

