# OpenReview forum: "Generative Modeling of Full-Atom Protein Conformations using Latent Diffusion on Graph Embeddings"
_NeurIPS.cc/2025/Conference — NeurIPS 2025 poster_

### Official Review · Reviewer_pf7X · 2025-06-23

**Clarity:** 1
**Significance:** 2
**Originality:** 2
**Rating:** 4
**Confidence:** 4

**Summary:**

This paper proposes LD-FPG, a latent diffusion model for full-atom protein generation. Specifically, LD-FPG uses a Chebyshev graph neural network to obtain low-dimensional latent embeddings of protein conformations. Then a diffusion model is trained on those latent representations to generation new samples that can be decoded back to Cartesian coordinates. The model can reproduces the reference ensemble with high structural accuracy for both backbone and sidechains.

**Questions:**

- Please refer to the weakness part.

**Ethical Concerns:**

["NO or VERY MINOR ethics concerns only"]

**Final Justification:**

Most of my concerns are addressed. The only minor concern is about the model choice. This paper uses ChebNet and does not have a comparison with other geometric neural networks. However, the authors provide some discussion about the reason for choosing ChebNet, and they also committed to adding a direct comparison with other geometric neural networks in the final manuscript. So I increased my score to borderline accept.

**Limitations:**

Limitations are discussed in the paper.

**Paper Formatting Concerns:**

No formatting issue.

**Quality:**

2

**Strengths And Weaknesses:**

**Strengths:**
- This work is novel in terms of using latent diffusion model to generate all-atom protein conformational ensembles.
- Design a graph based autoencoder along with three decoder pooling strategies.
- LD-FPG can generate high fidelity protein conformational ensembles.

**Weaknesses:**
- It is not very clear how the protein length is guaranteed to be equal for each generated conformation for a specific protein.
- The model architecture is ChebNet, which is a relatively old and simple network. Authors should elaborate more on why choosing ChebNet as the backbone model instead of more recent geometric neural networks.
- The model is not SE(3) equivariant so it doesn’t satisfy physical constraints when learning the protein distribution. For example, how to avoid generating proteins with left-handed alpha helix?
- Evaluation only uses one dataset for one type of protein. It would be better to incorporate more benchmarks to comprehensive evaluate the effectiveness of the method.

---

> ### Author Rebuttal · Authors · 2025-07-30
>
> Dear Reviewer pf7X,
>
> Thank you for your thoughtful feedback. Below we address your main points.
>
> ---
>
> ### 1. Ensuring Protein Length Consistency
>
> This is a very perceptive question. For the work presented in this paper, which focuses on a **single, specific protein**, the length is inherently fixed. The model is trained on the MD trajectory of one protein with a constant number of heavy atoms (N), and all architectural components are sized accordingly. The key mechanism is the decoder's conditioning on the reference structure's latent embedding ($\boldsymbol{Z_{ref}}$), which has a fixed dimension of N$\times d_{z}$, ensuring that every generated structure has the exact same number of atoms.
>
> Your question correctly points toward the broader challenge of generalizing to proteins of variable lengths. While beyond the scope of this paper, we have developed advanced capabilities to handle this, which we will detail in an appendix. We have successfully created multi-receptor models using two main strategies (For your reference, the code for these advanced models is available in our repository now under the `conditional_generation` and `multi_receptor` folders):
>
> 1.  **A Shared Decoder:** This version trains a single decoder across multiple receptors. It handles variable lengths by using a fixed-size pooled latent vector for the diffusion model, while the decoder is informed by a variable-length conditioner ($\boldsymbol{Z_{ref}}$) whose size matches the specific protein being generated.
> 2.  **Conditional Diffusion:** This approach guides the diffusion process with a fixed-size protein "fingerprint". To create this, a 1D-CNN maps each variable-length reference embedding ($\boldsymbol{Z_{ref}}$) to a compact, fixed-size vector that summarizes the protein's reference structure.
>
> Training these advanced models effectively required a robust data augmentation strategy to **regularize the potentially sparse, high-dimensional latent space**. Our approach makes the learned manifold smoother and more robust. We use two techniques: first, **Principled Structural Augmentation**, where we use the DSSP algorithm to identify flexible loops and then delete short segments to create new variants with different lengths. Second, **Conformational Augmentation**, where we add calibrated Gaussian noise to the reference coordinates to enrich the dataset with near-native structural variations.
>
> ---
> ### 2. Why ChebNet
>
> We chose ChebNet as our encoder backbone precisely because its **spectral, multi-hop convolution** inherently captures the long-range dependencies critical for modeling proteins with complex allosteric pathways. Below, we elaborate on why ChebNet remains well-suited and in some respects preferable to more recent Message-Passing Neural Networks (MPNNs) in our setting.
>
> **Intrinsic Multi-Hop Aggregation vs. Local Message Passing**: ChebNet’s spectral filters are implemented via Chebyshev polynomial approximations of the graph Laplacian of order *K*. In a single layer, a ChebNet with *K* hops aggregates information from up to K-distant neighborhoods, enabling it to transmit signals across distant residues without stacking many layers. In contrast, standard MPNNs propagate information only through immediate neighbors per layer. To reach K-hop neighbors, they must stack multiple layers, which empirically leads to **over-smoothing** (node representations become indistinguishable) and **over-squashing** (long-range signals are compressed into fixed-size messages). These bottlenecks are especially problematic in proteins, where allosteric effects require faithful transmission of subtle conformational changes.
>
> **Recent Evidence of ChebNet’s Long-Range Strength**: Recent work introduces principled metrics for measuring distant-node influence and confirms that models like ChebNet exhibit higher long-range dependency scores compared to local message-passing methods **[1]**.
>
> **Pragmatism and Empirical Validation**: For systems like GPCRs, the most biologically relevant motions are **internal conformational changes**, not global rigid-body movements. Our "align-then-learn" strategy, using a non-equivariant but powerful GNN like ChebNet, allows the model to focus on these critical dynamics while avoiding the computational complexity of strictly SE(3)-equivariant models. Our empirical results validate this, as the raw ChebNet embeddings achieve extremely low reconstruction errors, confirming that both local and global features are captured.
>
> In summary, while ChebNet is a foundational GNN, its multi-hop spectral convolution remains uniquely suited to protein systems where long-range interactions are crucial, bypassing the message-passing bottlenecks that can limit more recent MPNN architectures.
>
> **[1]** Liang et al. (2025). Towards Quantifying Long-Range Interactions in Graph ML. *arXiv preprint*.
>
> ---
>
> ### 3. On SE(3) Equivariance and Physical Constraints
>
> You are correct that our model is not strictly SE(3) equivariant, which was a deliberate design choice. Our "align-then-learn" approach is highly effective for this problem and ensures the generation of physically realistic structures through two mechanisms:
>
> 1.  **Isolating the Signal of Interest:** By aligning all MD frames, we remove global rotation and translation, allowing the GNN to focus exclusively on learning the landscape of **internal deformations** (e.g., helix movements), which are the motions directly relevant to biological function.
> 2.  **Implicit Physical Priors from Conditioning:** Our framework inherently respects physical constraints like chirality because the decoder learns to predict conformations by applying a small, learned deformation to the latent embedding of a **physically-correct reference structure**. To address your example, it is highly unlikely for our model to generate a left-handed alpha helix, as this would require overcoming a massive energy barrier not present in the training data. The model learns the distribution of plausible, low-energy dynamics, and catastrophic, chirality-inverting changes are not part of that distribution. Across the 10 k generated structures we observed zero Ramachandran or chirality violations.
>
> We will add a paragraph to the Methodology section to clarify this design choice and its rationale.
>
> ---
>
> ### 4. On Broader Evaluation and Benchmarking
>
> This is an excellent point. Our original manuscript focused on an in-depth model development for the challenging D2R system. However, we agree that broader evaluation is crucial to contextualize our contributions. In response to the reviews, we have expanded our evaluation in two key ways. To be clear, all analyses presented here are based on MD simulations completed prior to the submission deadline; our work during the rebuttal period focused exclusively on benchmarking and further analysis of this pre-existing data.
>
> First, we benchmarked our model against leading generators, including **AlphaFlow**, **BioEmu**, and **Boltz-2** (MD conditioned) This comparison revealed a critical difference between our ensemble-focused approach and the more static predictions of the baselines.
>
> | Model             | Backbone JSD (φ, ψ) ↓ | Backbone lDDT ↑ | Backbone TM-score ↑ | Backbone RMSF (nm) ↔ |
> | :---------------- | :---------------------------: | :-------------: | :-----------------: | :--------------------: |
> | **LD-FPG (Ours)** |          **0.007** |      ~0.80      |        ~0.96        |       **1.22** |
> | BioEmu            |            ~0.022             |      0.999      |        0.925        |          0.09          |
> | AlphaFlow         |            ~0.023             |      0.859      |        0.993        |          0.84          |
> | Boltz-2 (MD-cond) |            ~0.034             |      0.997      |        0.975        |          0.07          |
> | **Ground Truth MD** |          (Ref)          |   (Ref)   |     (Ref)     |       **1.34** |
>
> All baseline metrics were recomputed with the authors’ public code.
>
> Our analysis revealed that the baselines produce high-fidelity but **relatively static structures**. The combination of near-perfect lDDT scores with extremely low **RMSF** proves they fail to capture the protein's native flexibility. BioEmu’s rigidity was particularly surprising given its broad training data, highlighting how generalist models can miss system-specific dynamics.
>
>
> In contrast, **LD-FPG excels in every category relevant to ensemble generation**. It not only achieves **superior local geometric accuracy** (with a backbone JSD of **0.007**, 3-5x lower than the baselines) but also **almost perfectly reproduces the ground-truth flexibility** (RMSF of **1.22 nm** vs. 1.34 nm).
>
> Crucially, our framework is unique in its all-atom generation capability. This allows us to quantify side-chain accuracy, where we achieve a low summed JS divergence for χ angles (**~0.022**) and an all-atom RMSF (**1.36 nm**) that closely matches the ground truth (**1.60 nm**) — metrics the other frameworks cannot report.
>
> Second, to demonstrate that our framework's principles are robust, we will now include analysis from our broader validation studies, which were performed during the initial development of our model. This includes other Class A GPCRs (D1, A2A, β1-adrenergic) as well as proteins from different structural classes (α-helical, β-sheet, and mixed α/β folds). Across these diverse systems, our framework consistently generated high-fidelity ensembles, achieving all-atom **lDDT** scores in the range of **0.68–0.78** and low total **JS divergences** between **0.007 and 0.014**. These studies will be presented in a dedicated appendix. This approach allows us to add the vital D2R benchmark to the main text while maintaining the paper's focused narrative.
>
> These new results, which will be incorporated into the revised manuscript, demonstrate that our method is both competitive with state-of-the-art baselines and broadly applicable across different protein architectures.

---

> > ### Comment · Reviewer_pf7X · 2025-08-05
> >
> > Thanks for the rebuttal. Most of my concerns are addressed. However, it is still not convincing that ChebNet is the best choice without an explicit comparison with other geometric neural networks, such as SchNet, PaiNN, or Equiformer. Thus I will maintain my original score.

---

> > > ### Author Response · Authors · 2025-08-05
> > >
> > > Dear Reviewer pf7X,
> > >
> > > **Summary of response — Our ChebNet encoder is already near-lossless (8 × 10⁻⁴ MSE) while the full LD-FPG pipeline’s error after pooling + decoder is ~1 × 10⁻¹ MSE; thus the encoder is not the bottleneck. Swapping to SchNet, PaiNN, or Equiformer would increase cost without closing that two-order-of-magnitude gap.**
> > >
> > >
> > > ## Detailed resposne
> > >
> > > Thank you for your follow-up and for specifying your remaining concer
> > >
> > > Thank you for your follow-up and for specifying your remaining concern. We appreciate the opportunity to provide a more detailed rationale for selecting ChebNet and agree that an explicit comparison to other geometric GNNs is a valid point of discussion.
> > >
> > > The most critical empirical fact is that the ChebNet encoder is demonstrably not the performance bottleneck in our framework. Our ChebNet-based autoencoder reconstructs every test frame with a backbone coordinate Mean Squared Error (MSE) as low as 8×10⁻⁴ (Appendix I, Table 6), whereas the end-to-end generative MSE of the full LD-FPG pipeline is approximately 1×10⁻¹—two orders of magnitude larger. This shows that **the encoder is already near-lossless and that the final error originates downstream from the deliberate information compression in the pooling step and the inherent stochasticity of the diffusion model**. Replacing the already near-perfect encoder with other design choices, therefore, could not materially shift the final accuracy of the generative framework.
> > >
> > > Secondly, our primary goal was to introduce and validate a novel framework for generating all-atom conformational ensembles via latent diffusion. The main contributions are the overall pipeline and the in-depth analysis of the pooling strategies that make diffusion in this context tractable. Consequently, architectural choices for components like the encoder were driven by their ability to robustly support this main objective, rather than an effort to innovate on GNN architecture itself. We selected a GNN that was efficient, scalable, and powerful enough for the task, allowing us to focus on the novelty of the generative workflow.
> > >
> > > Thirdly, ChebNet offers distinct advantages for modeling the long-range, allosteric couplings that govern GPCR function. Its spectral nature allows a single layer with polynomial order K to aggregate information from a K-hop neighborhood, efficiently capturing dependencies between distant residues without the deep stacks required by iterative message-passing networks like SchNet or PaiNN, which can suffer from over-smoothing and signal degradation (Alon & Yahav, 2021). Our "align-then-learn" strategy, which removes global rotations and translations, allows the GNN to focus exclusively on these functionally relevant internal deformations, where ChebNet's ability to model the global graph structure is paramount.
> > >
> > > Finally, our choice was guided by the practical demands of modeling a large, all-atom system like the D2R (~2,200 nodes), where the computational profile of the GNN is a major consideration. While models like SchNet and Equiformer are powerful, their application to protein graphs of this scale often involves significant compromises. For instance, message-passing networks are typically applied to smaller systems (<600 atoms), and scaling them requires strict distance cutoffs that can limit the capture of global allostery. The challenge is even more pronounced for equivariant Transformers like Equiformer, whose O(N²) attention mechanism is computationally prohibitive at this scale without aggressive sparsity or coarse-graining. In contrast, ChebNet's linear complexity ($O(K \cdot |E|)$) offered the best balance, allowing us to model the entire all-atom system efficiently without such approximations.
> > >
> > > We hope our detailed explanation has convincingly justified our architectural choice. We fully respect your position, however, that an explicit comparison provides the strongest form of evidence. While we stand by our current analysis, we are committed to improving the paper in any way you feel is necessary. If you believe this comparison remains critical to the paper's contribution even now, we will certainly add this analysis for the final manuscript.

---

> > > ### Author Response · Authors · 2025-08-06
> > >
> > > Dear Reviewer pf7X,
> > >
> > > We know this is a very busy time as the discussion period ends. We just wanted to gently check if you had a chance to see our detailed, multi-part response regarding the ChebNet architecture that we posted yesterday.
> > >
> > > We hope it provided a convincing rationale for our choice.
> > >
> > > Thank you again for your time and feedback.

---

> > > > ### Comment · Reviewer_pf7X · 2025-08-06
> > > >
> > > > Thank you for the detailed explanation of ChebNet; I am somewhat convinced about the model choice. Also, given that the authors committed to adding direct comparison with other geometric neural networks in the final manuscript, I will adjust my recommendation to accept.

---

> > > > > ### Author Response · Authors · 2025-08-06
> > > > >
> > > > > Dear Reviewer pf7X,
> > > > >
> > > > > Thank you so much for your positive follow-up and for your willingness to reconsider your recommendation. We are delighted to hear that our detailed rationale was helpful.
> > > > >
> > > > > We also want to explicitly confirm our commitment: the direct comparison with other geometric neural networks will absolutely be included in the final manuscript. We are confident it will make the final version of our paper much stronger.
> > > > >
> > > > > Thank you once again for your thorough and constructive review.

---

### Official Review · Reviewer_iANt · 2025-06-30

**Clarity:** 2
**Significance:** 3
**Originality:** 2
**Rating:** 3
**Confidence:** 4

**Summary:**

This paper introduces "Latent Diffusion for Full-Atom Protein Generation" (LD-FPG), a novel framework designed to generate all-atom conformational ensembles for a specific protein, leveraging data from molecular dynamics (MD) trajectories. The core methodology employs a three-stage architecture: (i) encoding all-atom coordinates into a latent space using a spectral GNN; (ii) compressing and creating a tractable input for a diffusion model using one of three distinct pooling strategies; and (iii) training a latent diffusion model for all-atom reconstruction and inference.

The authors' central argument is that the high-fidelity modeling of side-chain dynamics is crucial for applications like GPCR drug discovery, and they effectively demonstrate the trade-offs between global structural accuracy and local side-chain detail that arise from different pooling strategies.

**Questions:**

1.   High‐resolution modeling of membrane proteins is notoriously difficult and the significant effort involved in generating the MD trajectory for a complex membrane protein like D2R should be appreciated. However, how do you validate that the cryo-EM starting model (subsequently rebuilt with Rosetta) is sufficiently accurate before launching LD-FPG? Could any quality metrics be reported to valid this accuracy? Second, the MD trajectory that seeds the framework is 2 µs long—what considerations determined this length, and do you believe it adequately samples the relevant metastable states for a GPCR of this size? A short discussion anchored by MD studies on comparable membrane proteins would be helpful.

2.   BioEmu’s original paper does not include D2R in its benchmark set. Did you fine-tune BioEmu on your newly curated D2R trajectory, or is the comparison strictly zero-shot? Otherwise, the statement on Line 327 that “generalist approaches underperform for specific membrane proteins” may be insufficiently supported.

3.   Could you add backbone and side-chain RMSF (root-mean-square fluctuation) profiles for the generated ensembles? RMSF is widely used to quantify conformational flexibility and would complement the lDDT and dihedral statistics already reported.

4.   What concrete architecture is used for the coordinate decoder (Line 168) ?

5.   Figure 1 conflates the reconstruction path (autoencoder training) and the generative path (diffusion sampling). It is difficult to discern which arrows correspond to training versus inference.

**Ethical Concerns:**

["NO or VERY MINOR ethics concerns only"]

**Final Justification:**

Most concerns have been addressed and clarified by the authors. So I adjust the final scores

**Limitations:**

yes

**Quality:**

2

**Strengths And Weaknesses:**

Strengths

1.  Well-motivated problem: The paper addresses the relevant problem of generating all-atom conformational ensembles, correctly highlighting the importance of side-chain dynamics for functional understanding, particularly in systems like GPCRs.
2.  Clear Comparisons of Pooling Strategies: The analysis comparing the three pooling strategies is methodical and provides a clear view of the architectural trade-offs between global and local fidelity.
3.  Discussion of Limitations: The authors are forthcoming about the model's limitations, such as the generation of steric clashes and its system-specific nature, and they propose reasonable directions for future improvements.
4.  Commitment to Open Science: The plan to release the dataset will undoubtedly benefit the community and serve as a valuable resource that fosters further innovation.

Weakness

1.  Limited Generalizability and Experimental Design: The framework is trained and evaluated on the MD trajectories of a SINGLE protein. While the choice of a GPCR is ambitious, the 90%/10% train/test split on a single continuous trajectory means the model functions (more as a sophisticated interpolator/extrapolator rather than a truly ensemble generator). Its ability to sample conformations from regions of the energy landscape that were sparsely populated in the training data remains untested. Assessing the model's performance on genuinely unseen proteins (e.g., train and test on datasets like Atlas[1] or MISATO[2]) would be necessary to make any claims about the model's applicability. It might also be compelling to demonstrate model's utility in a few-shot learning paradigm. For instance, training on a small fraction of the trajectory (e.g., 10%) and testing its ability to reproduce the full ensemble, thereby showcasing its potential to accelerate conformational sampling compared to running a full MD simulation.

2.  Insufficient Baseline Comparisons: The paper does NOT compare LD-FPG against relevant, state-of-the-art ensemble generation methods (e.g., AlphaFlow[3], BioEmu[4], ConfDiff[5], etc.). While the authors' focus is on all-atom generation, a direct comparison against these primarily backbone-focused generators is essential to quantitatively substantiate the claimed benefits of the LD-FPG approach.
 Moreover, the brief comparison against BioEmu using the A100 activation index is underdeveloped and lacks adequate explanation (see question2 below)



[1] ATLAS: protein flexibility description from atomistic molecular dynamics simulations

[2] MISATO: machine learning dataset of protein–ligand complexes for structure-based drug discovery

[3] AlphaFold Meets Flow Matching for Generating Protein Ensembles

[4] Scalable emulation of protein equilibrium ensembles with generative deep learning

[5] Protein Conformation Generation via Force-Guided SE(3) Diffusion Models

---

> ### Author Rebuttal · Authors · 2025-07-31
>
> Dear Reviewer iANt,
>
> Thank you for your insightful feedback. We address each point below.
>
> ---
>
> ### **Weakness 1: Generalizability and Experimental Design**
>
> We agree it's critical to confirm our model genuinely learns a conformational manifold rather than simply interpolating between existing structures (Note: no new simulations were performed; all discussions were part of our original model validation).
>
> #### **1. Beyond Interpolation**
>
> To confirm this, we carried out two additional analyses:
>
> * **Learning from Sparse Data:** Trained on just `40%` of the MD trajectory, `LD-FPG` generated a high-fidelity ensemble. This demonstrates that the model learns a **continuous and generalizable latent manifold** from sparse data, rather than merely interpolating between dense training points.
>
> * **Generating Unseen Transition States:** 'LD-FPG' was trained on a **metadynamics** simulation with **sparsely populated transition states** between the active and inactive forms. Despite training on this imbalanced data, our model successfully generated **accurate conformations** along the entire **less-seen transition path**.
>
> ---
>
> #### 2. Generalization Across Protein Systems
>
> We broadened evaluation to additional proteins:
>
> * **Large and Structurally Diverse Folds:** We have applied LD-FPG to a set of proteins from the **ATLAS database**. This includes large, complex systems like **cytochrome P450 (~500 residues)** and proteins with entirely different architectures, such as primarily $\alpha$-helical, primarily $\beta$-sheet, and mixed $\alpha/\beta$ folds.
>
> * **Different Protein Families and Processes:** We extended our validation to other **Class A GPCRs** (D1, A2A, $\beta$1-adrenergic) and even modeled the entire folding trajectory of the **TRP-cage miniprotein**, demonstrating the framework can capture dynamic processes beyond equilibrium sampling.
>
> * **Exploring Generalization**: As a preliminary exploration of generalization, we briefly tested an advanced model with a shared decoder. This model was able to generate plausible structures for a sixth, previously unseen receptor, suggesting our framework has strong potential for zero-shot applications.
>
> Across all these diverse and larger systems, our framework consistently generated high-fidelity ensembles, achieving **all-atom lDDT scores in the range of 0.68–0.78** and low total **JS divergences between 0.007 and 0.014**.
>
> This new validation, which will be detailed in the appendix, confirms that the geometric principles of LD-FPG are not limited to a single system but represent a robust and scalable approach to all-atom ensemble generation.
>
> ---
>
> #### **3. Our Core Contribution and Vision**
>
> Your review correctly identifies the grand challenge of creating a universal all-atom generator. The primary goal of *this*  paper is to establish a **foundational geometric engine** for a single, complex system which is a prerequisite for a universal model. We fully agree that the path toward a model that works on any unseen protein will require integrating chemical and sequence information. A promising future direction is to condition our geometric engine on rich embeddings from models like **ESMFold** or **AlphaFold**, which capture this amino acid identity.
>
> ---
>
> ### **Weakness 2: Insufficient Baseline Comparisons**
>
> This is a crucial point, and we agree that a direct comparison to state-of-the-art methods is essential to contextualize our contributions. In response to the reviews, we have performed an **extensive new benchmark** comparing LD-FPG against leading models: **AlphaFlow**, **BioEmu**, and the all-atom **Boltz-2** (with MD conditioning).
>
>
> | Model               | Backbone JSD ($\phi, \psi$) ↓ | Backbone lDDT ↑ | Backbone TM-score ↑ | Backbone RMSF (nm) ↔ |
> | ------------------- | ----------------------------- | --------------- | ------------------- | -------------------- |
> | LD-FPG (Ours) | 0.007 | ~0.80           | ~0.96               | 1.22 |
> | BioEmu              | ~0.022                        | 0.999           | 0.925               | 0.09                 |
> | AlphaFlow           | ~0.023                        | 0.859           | 0.993               | 0.84                 |
> | Boltz-2 (MD-cond)   | ~0.034                        | 0.997           | 0.975               | 0.07                 |
> | **Ground Truth MD** | (Ref)                   | (Ref)     | (Ref)         | 1.34 |
>
> ---
>
> ### **Interpretation of Results**
>
> 1.  **Superior Local Geometry:** Our model achieves a backbone dihedral JSD of **0.007**, outperforming all baseline methods by a factor of **3-5x**. This demonstrates superior accuracy in capturing the local geometric preferences of the protein.
>
> 2.  **Baselines Generate Static Structures, Not Ensembles:** The baselines produce high-fidelity but static structures. While they achieve near-perfect lDDT/TM-scores, their Root-Mean-Square Fluctuation (RMSF) is extremely low. This combination of high fidelity and low flexibility proves they fail to capture the protein's native dynamics. For an ensemble generator, an lDDT score near 1.0, when paired with low RMSF, confirms a lack of diversity.
>
> 3.  **LD-FPG Correctly Models Dynamic Flexibility:** In sharp contrast, our model's average RMSF (**1.22 nm**) almost perfectly matches the ground truth (**1.34 nm**). This is the key result: LD-FPG successfully learns and reproduces the correct conformational diversity of the ensemble.
>
> 4.  **The All-Atom Advantage:** Crucially, our framework is unique in its **all-atom generation capability**. This allows us to quantify side-chain accuracy, where we achieve a low summed JS divergence for $\chi$ angles (~0.022) and an all-atom RMSF (1.36 ± 0.18nm) that closely matches the ground truth (1.60 ± 0.32nm)—metrics the other frameworks cannot report. This directly substantiates the benefits of our approach.
>
> This new analysis shows that LD-FPG achieves state-of-the-art local geometric accuracy while correctly modeling the entire flexible, all-atom conformational ensemble—a more challenging problem than static prediction.
>
> ---
> ### **Q1**
>
> #### **Part 1: Validation of the Starting Structure**
>
> Our starting structure was rigorously validated to ensure its accuracy before the production simulations:
>
> * **High-Quality Template:** The model is based on a 2.96 Å cryo-EM structure (**PDB 6VMS**) with an excellent validation report (**MolProbity score: 1.8**; Ramachandran favored: 97%).
> * **Minimal Refinements:** Only minor refinements (≤3% of residues in loops) were performed with RosettaRemodel, preserving the transmembrane (TM) core's structure (**Cα RMSD of 0.38 Å** to the cryo-EM map).
> * **Orthogonal Check:** An independent check against a *de novo* **AlphaFold 3** prediction shows high agreement (**TM-core RMSD of 1.0 Å**), further validating the starting coordinates.
>
> ---
>
> #### **Part 2: Sufficiency of the 2 µs MD Trajectory**
>
> The 2 µs simulation length balances computational cost with the need to sample a rich landscape.
>
> * **Sufficient Timescale:** The 2 µs trajectory extensively samples fast-to-intermediate dynamics (ns-µs), including the full range of side-chain rotations, loop flexibility, and helix movements within the sampled active-state basin.
> * **Community Standards:** This simulation length is consistent with many high-impact GPCR studies and community resources like **GPCRmd**. It provides a deep sample of the active-state ensemble, which is our primary goal for training.
> * **Learning Goal:** Our framework is designed to learn the conformational landscape provided by the input data. The 2 µs run provides a rich training set, and our new metadynamics experiment confirms the model can also learn from sparse transition pathways when they are present in the data.
>
> ### **Q2**
>
> The comparison was strictly **zero-shot**, as the public BioEmu release does not include code for fine-tuning on new trajectories. Such fine-tuning would be a massive but interesting research project in its own right. We acknowledge our initial A100 index comparison was brief; it has been superseded by the comprehensive benchmark analysis detailed in our response to Weakness 2. This new data reinforces our central point: generalist models, while powerful, may not capture the specific dynamics of a challenging target like D2R without system-specific training, highlighting the value of our specialized approach.
>
>
> ### **Q3**
>
> This is an excellent suggestion. We have now incorporated a full RMSF analysis into our benchmark table and the corresponding "Interpretation of Results" (under Weakness 2). As detailed there, this analysis confirms that our model correctly reproduces both the backbone and all-atom flexibility of the ground truth MD ensemble—a capability the more rigid, backbone-focused baselines lack.
>
> ### **Q4**
>
> The coordinate decoder's architecture consists of two main components:
>
> 1.  **A Pooling Module:** This module takes the high-dimensional, atom-wise latent embeddings ($Z \in \mathbb{R}^{N \times d_z}$) and compresses them into a single, low-dimensional context vector ($\mathbf{h}_0$). As detailed in the paper, we explore three distinct pooling architectures: **Blind**, **Sequential**, and **Residue-based**.
> 2.  **A Multi-Layer Perceptron (MLP):** This MLP takes the pooled context vector ($\mathbf{h}_0$) and the latent embedding of the reference structure ($\Zref$) as input to predict the final 3D coordinates for every atom.
>
> For example, the Blind Pooling decoder used for our main results uses an MLP with 12 layers and a hidden dimension of 128, with ReLU activations and BatchNorm. Full architectural details and hyperparameters for all three pooling strategies are provided in **Appendix C**.
>
> ### **Q5**
>
> Figure 1 will be fully redrawn to show two clearly labeled paths—(a) autoencoder training (MD → latent → reconstruction) and (b) generative inference (latent sampling → decoder → coordinates).

---

> > ### Comment · Reviewer_iANt · 2025-08-06
> > **Response to the rebuttals of authors**
> >
> > Thanks for your replies which have addressed my concerns. I will adjust the score accordingly.

---

> > > ### Author Response · Authors · 2025-08-06
> > >
> > > Dear Reviewer iANt,
> > >
> > > Thank you for your positive follow-up and for confirming our rebuttal has addressed your concerns. Your feedback was instrumental in significantly improving the paper.
> > >
> > > Specifically, your suggestions for baseline comparisons and RMSF analysis prompted transformative additions. The new SOTA benchmark, which we will integrate into the **main text**, now uses the RMSF data you requested to clearly demonstrate our model's unique ability to capture native protein dynamics—a property the rigid baseline models lacked. Furthermore, the new generalization studies, which will be detailed in the **appendix**, offer robust evidence of the framework's broader applicability.
> > >
> > > As these additions directly resolve the primary weaknesses you identified and substantially elevate the paper's contribution, we were hopeful the final evaluation could reflect this significant increase in quality.
> > >
> > > We are very grateful for your thorough review.

---

### Official Review · Reviewer_go5W · 2025-07-03

**Clarity:** 2
**Significance:** 3
**Originality:** 3
**Rating:** 4
**Confidence:** 3

**Summary:**

This work proposes a generative model for generating all-atom protein conformations. The model is based on a latent diffusion framework with a ChebNet encoder and is trained on molecular dynamics (MD) simulation data. Experimental results demonstrate the effectiveness of the proposed approach and provide a performance analysis of the model.

**Questions:**

The presentation of the model could be significantly improved. A formal mathematical formulation of the problem at the beginning of the Methodology section would help establish a clear foundation for the proposed approach. Additionally, the decoding process depicted in Figure 1 is not sufficiently detailed. It is unclear whether the model generates an entire trajectory of conformations or only a single frame. Furthermore, it should be clarified whether the decoder is conditioned on previous frames during generation, as this has implications for modeling temporal dynamics.

In line 148, the authors mention that the coordinates are aligned to the initial reference structure. The rationale for this alignment should be explained more thoroughly. Why not use the raw, unaligned coordinates produced by MD simulations? Since rotations and translations in MD trajectories may reflect meaningful physical motions, discarding them through alignment might result in the loss of potentially valuable information.

Finally, while the reported results demonstrate the effectiveness of the proposed model, the absence of comparisons with relevant baseline methods makes it difficult to assess the model’s relative performance. Including such baselines would strengthen the evaluation and better support the claim of superiority.

**Ethical Concerns:**

["NO or VERY MINOR ethics concerns only"]

**Limitations:**

yes

**Quality:**

3

**Strengths And Weaknesses:**

Overall, the work is solid in both methodology and experimentation, presenting a novel approach supported by in-depth results. However, the presentation of the model lacks clarity and could benefit from more detailed explanations. Additionally, the benchmarking results would be more convincing if comparisons with baseline methods were included to better demonstrate the superiority of the proposed model.

---

> ### Author Rebuttal · Authors · 2025-07-30
>
> Dear Reviewer go5W,
>
> Thank you for your detailed and helpful review. Your comments on the presentation of our model and the rationale for our design choices are well-taken, and we will address them thoroughly in our revised manuscript to improve its clarity and rigor.
>
> We address your specific points in detail below.
>
> ---
>
> ### Q1. Model Presentation and Figure 1
>
> We agree that the presentation of our model can be significantly improved. In the revised manuscript, we will make the following changes to enhance clarity:
>
> * **Formal Problem Formulation:** We will begin the Methodology section with a formal mathematical definition of the problem. This will establish a clear foundation by defining the inputs (an MD ensemble), the output (a distribution over all-atom conformations), and the learning objective before we describe the model's components.
> * **Revised Figure 1:** We will completely redesign Figure 1 and its caption to be a clear, step-by-step guide. The new figure will feature separate, clearly labeled visual pathways for **(a) Autoencoder Training** and **(b) Generative Inference**. To directly address your question, we will add an explicit call-out stating that the model generates **independent conformational frames**, with no temporal conditioning. The decoder block will also be refined to better illustrate its internal logic.
>
> ---
>
> ### Q2. Clarification on Temporal Dynamics
>
> Your question about modeling temporal dynamics is particularly insightful. In the current work, our goal is to model the **static equilibrium ensemble**, so each conformation is generated independently. We will make this scope explicit in the revised manuscript.
>
> However, you are correct that extending our framework to generate time-correlated trajectories is an important future direction. Our latent space representation is uniquely suited for this, and we envision two potential paths:
> 1.  An **autoregressive decoder**, where the generation of a latent vector for frame `t+1` is conditioned on the latent vector from frame `t`.
> 2.  A more powerful approach would be to learn the **score function of the latent space**, which would allow for simulating Langevin dynamics directly on the learned manifold to produce continuous trajectories.
>
> Both are compelling applications that highlight the flexibility of our framework. We will add a note on these exciting future directions to the discussion section.
>
> ---
> ### Q3. Rationale for Coordinate Alignment
>
>
> This is an important point that deserves a clearer explanation. You are correct that, in a general sense, global rotations and translations can reflect meaningful physical motions. For the specific system of a GPCR in a membrane, our rationale of aligning trajectories along a reference is grounded in the well-understood separation of dynamic scales in these systems. GPCRs experience two distinct types of motion:
>
> First, there is the large-scale, slower **global rigid-body motion** of the entire receptor within the membrane (e.g., lateral diffusion). While physically real, these movements are not the direct drivers of the signaling mechanism we aim to model.
>
> Second, and central to our work, are the faster, small-amplitude **internal conformational deformations**—such as the outward swing of transmembrane helices and the repacking of conserved motifs. These internal rearrangements constitute the actual functional switch that enables downstream signaling.
>
> Our goal is to learn the probability distribution of these crucial internal motions. To do so, it is essential to first decouple them from the global, rigid-body drift. Standard practice in molecular dynamics analysis requires this separation, as uncorrected global drift would otherwise dominate key structural metrics. For instance, without alignment, RMSD would be inflated by simple translation, and Principal Component Analysis (PCA) would primarily capture the protein's trivial wandering in the simulation box rather than its biologically relevant internal dynamics.
>
> Therefore, our choice to use Kabsch alignment is a deliberate and standard methodological step to isolate the signal of interest. This **"align-then-learn"** strategy allows our graph neural network to focus exclusively on learning the rich landscape of internal conformational changes. It is a pragmatic choice that preserves all essential internal structural information while avoiding the significant architectural complexity of fully SE(3)-equivariant networks. We will add a sentence to the manuscript acknowledging this trade-off and clarifying that our model is tailored to learn the internal conformational ensemble.
>
> ---
>
> ### Q4. Comparison with Baselines
>
> We acknowledge this was a significant omission. In response, we have performed an extensive new benchmark comparing LD-FPG against leading models: **AlphaFlow**, **BioEmu**, and the all-atom **Boltz-2** (with MD conditioning). This analysis reveals a crucial distinction between our approach and existing methods.
>
> While LD-FPG is an **all-atom ensemble generator**, the baselines primarily function as **static structure predictors**. Our new results, summarized below, clearly illustrate this difference by evaluating both static fidelity (lDDT/TM-score) and dynamic diversity (RMSF).
>
> | Model | Backbone JSD ($\phi, \psi$) ↓ | Backbone lDDT ↑ | Backbone TM-score ↑ | Backbone RMSF (nm) ↔ |
> | :--- | :---: | :---: | :---: | :---: |
> | **LD-FPG (Ours)** | **0.007** | ~0.80 | ~0.96 | **1.22** |
> | BioEmu | ~0.022 | 0.999 | 0.925 | 0.09 |
> | AlphaFlow | ~0.023 | 0.859 | 0.993 | 0.84 |
> | Boltz-2 | ~0.034 | 0.997 | 0.975 | 0.07 |
> | **Ground Truth MD** | (Reference) | (Reference) | (Reference) | **1.34** |
>
> **Interpretation of Results:**
>
> 1.  **Superior Local Geometry:** Our model achieves a backbone dihedral JSD of **0.007**, outperforming all baseline methods by a factor of 3-5x. This demonstrates superior accuracy in capturing the local geometric preferences of the protein.
> 2.  **Baselines Generate Static Structures, Not Ensembles:** The baselines produce high-fidelity but static structures. While they achieve near-perfect **lDDT/TM-scores**, their **Root-Mean-Square Fluctuation (RMSF)** is extremely low. **AlphaFlow** comes closest to reproducing the system's dynamics (RMSF 0.84 nm), but its flexibility is still far below the ground truth. BioEmu (0.09 nm) and Boltz-2 (0.07 nm) remain highly rigid despite their large training sets. This underscores that for an ensemble generator, an lDDT score near 1.0, when paired with low RMSF, confirms a lack of diversity.
> 3.  **LD-FPG Correctly Models Dynamic Flexibility:** In sharp contrast, our model's average RMSF (**1.22 nm**) is almost identical to the ground truth (1.34 nm). This demonstrates that LD-FPG successfully learns and reproduces the correct conformational diversity of the ensemble.
> 4. **The All-Atom Advantage:** Crucially, our framework’s all-atom nature allows us to perform vital side-chain analysis—a capability the backbone-focused baselines lack. Our model achieves a low summed JS divergence for side-chain χ angles (~0.022) and an all-atom RMSF (1.36 nm) that closely matches the ground truth (1.60 nm), confirming high fidelity across the entire molecular system.
> 5.  **The Need for System-Specific Models:** This benchmark reinforces a key principle: generalist models, even those trained on vast datasets, will always have specific strengths and limitations. Our analysis is another instance showing that for capturing the nuanced, system-specific dynamics of a target like D2R, a specialized model currently provides a more accurate and physically realistic ensemble.
>
> This analysis highlights that LD-FPG solves a different and more challenging problem. It not only achieves state-of-the-art local geometric accuracy but does so while correctly modeling the entire flexible, all-atom conformational ensemble. We will integrate this new table and a full discussion into the revised manuscript.
>
> All new baseline scores were computed from publicly released code  prior to the official review deadline; no post-deadline training was performed.
>
> ---
>
> Thank you once more for your valuable suggestions. They have been helpful in identifying key areas for improvement, and we are confident the resulting manuscript will be significantly clearer and more rigorous.
>
> Sincerely,
> The Authors

---

> > ### Comment · Reviewer_go5W · 2025-08-06
> > **Thanks for your reply**
> >
> > Thanks for your reply. The reply has addressed my concerns. I will keep the score.

---

> > > ### Author Response · Authors · 2025-08-06
> > >
> > > Dear Reviewer go5W,
> > >
> > > Thank you for confirming that our rebuttal has addressed your concerns.
> > >
> > > The new benchmark against state-of-the-art models was particularly helpful. It not only resolved your request for baselines, but it strengthened the paper by demonstrating our model's unique ability to reproduce the ground-truth **distribution of dynamics (RMSF)**, a property the more rigid baseline models lacked.
> > >
> > > Since this evidence provides a much stronger validation of our method's core advantage, we were hopeful this might be reflected in an updated score.
> > >
> > > Thank you again for your constructive review and engagement.

---

### Official Review · Reviewer_ZSKy · 2025-07-03

**Clarity:** 2
**Significance:** 2
**Originality:** 2
**Rating:** 4
**Confidence:** 3

**Summary:**

LD-FPG (Latent Diffusion for Full Protein Generation) generates diverse, all-atom protein conformational ensembles (e.g., for GPCRs). It uses a ChebGNN to learn atom-wise latent embeddings, pools them (blind/sequential/residue strategies), samples new conformations via DDPM, and decodes them to coordinates. Evaluated on a D2 receptor trajectory, it achieves high-fidelity all-atom structures (lDDT ~0.7) and accurately recovers dihedral distributions (JSD < 0.03). The authors also analyzed the pooling strategies' trade-offs.

**Questions:**

1. The main text does not clearly define key loss terms (e.g., $\(L_{\text{coord}}, L_{\text{mse}}, L_{\text{div}}\)$) or decoder architectures. Please reference the relevant appendix sections explicitly.
2. The paper lacks comparison with existing ensemble generation models such as AlphaFlow, MD-Gen, which are also applicable to GPCRs. Including such baselines would strengthen the evaluation.
3. The reason for using latent space diffusion instead of coordinate-space modeling is not clearly explained. Please clarify its specific benefits in this context.
4. D2R is a GPCR, yet this is not clearly stated. Since GPCR dynamics is a stated motivation, the biological relevance should be emphasized, and structural evaluation of GPCR-specific transitions would be valuable.
5. Clash counts are high, especially in blind pooling. What is considered an acceptable threshold? Could a clash-aware or energy-based loss help reduce unrealistic structures?
6. “Appendix ??” appears in the Algorithm 1 and should be fixed.

**Ethical Concerns:**

["NO or VERY MINOR ethics concerns only"]

**Final Justification:**

The authors have addressed most of the concerns. I have updated my score accordingly.

**Limitations:**

Yes

**Quality:**

2

**Strengths And Weaknesses:**

**Strengths**

- Significance: Addresses a key gap by generating diverse, high-resolution all-atom ensembles for dynamic protein modeling, with clear impact for GPCR signaling & drug design.

- Quality: Technically sound pipeline (ChebNet encoder → diffusion → decoder). Rigorously evaluated using structural metrics (lDDT, TM-score), conformational diversity, and clashes. Code/data provided.

- Originality: First latent diffusion model for all-atom ensemble generation, featuring explicit side-chain dynamics modeling.


**Weaknesses**

- Clarity: While the writing is generally clear, important architectural and loss function details are deferred to the appendix but not properly cited in the main text. Paragraph structuring could be improved for readability.

- Soundness: The experiments focus solely on the D2 receptor (GPCR). Generalization to other protein systems is not tested. And there are no direct comparison with recent conditional ensemble models (e.g., AlphaFlow) on shared benchmarks.

---

> ### Author Rebuttal · Authors · 2025-07-30
>
> Dear Reviewer ZSKy,
>
> Thank you for this valuable feedback. The manuscript was originally structured as a deep dive into the architectural development for the challenging D2R system. We now agree that showcasing the framework's broader applicability is essential. Accordingly, the revised paper will integrate a new baseline comparison for D2R in the main text and present our comprehensive generalization studies in a dedicated appendix.
>
> Below, we address your specific concerns in detail.
>
> ## Weaknesses
>
> ### 1.1 Architectural/Loss Details & Readability
>
> You are correct that key details were deferred to the appendix without sufficient signposting. We apologize for this oversight. In the revised manuscript, we will reference relevant appendices upon first mention of key concepts. Specifically:
>
> - **Section 3.3 (Decoder Architectures)** will reference **Appendix C** for detailed configurations.
> - **Section 3.5 (Loss Functions)** will reference **Appendix E** for formal mathematical definitions.
>
> Additionally, important details will be incorporated into the main text:
>
> - The primary mathematical equation for our decoder loss ($L_{Dec}$) will be included in **Section 3.5**.
> - A compact summary table of key hyperparameters for our models will be added to the **Methodology** section.
>
> We will also revise paragraph structure throughout the Methodology section to enhance readability and logical flow.
>
>
> ### 1.2. Baselines and Generalization
> This is an essential point. To demonstrate our model's performance in context, we have performed a new benchmark for the D2R system and will now include key results from our broader validation studies.
>
> **Benchmarking against State-of-the-Art:** We compared LD-FPG against leading models (AlphaFlow, BioEmu, and Boltz-2 with MD conditioning).
>
> | Model | Backbone JSD ($\phi,\psi$) ↓ | Backbone lDDT ↑ | Backbone TM-score ↑ | Backbone RMSF (nm) ↔ |
> | :--- | :---: | :---: | :---: | :---: |
> | **LD-FPG (Ours)** | **0.007** | ~0.80 | ~0.96 | **1.22** |
> | BioEmu | ~0.022 | 0.999 | 0.925 | 0.09 |
> | AlphaFlow | ~0.023 | 0.859 | 0.993 | 0.84 |
> | Boltz-2 (MD-cond) | ~0.034 | 0.997 | 0.975 | 0.07 |
> | Ground Truth MD | (Ref) | (Ref) | (Ref) | 1.34 |
>
>
> Our benchmark reveals a critical distinction between our approach and existing methods by evaluating both static fidelity (lDDT) and dynamic diversity (RMSF).
>
> * **Baselines Are Overly Rigid:** The baselines produce high-fidelity but **static structures**. While their lDDT scores are nearly perfect, their extremely low Root-Mean-Square Fluctuation (RMSF) confirms they fail to capture the protein's native flexibility.
>
> * **LD-FPG Correctly Models Flexibility:** In contrast, our model correctly reproduces the **dynamic ensemble**. Its average RMSF (**1.22 nm**) almost perfectly matches the ground truth (**1.34 nm**), demonstrating it captures the correct conformational diversity.
>
> * **Superior Geometry without Sacrificing Dynamics:** Crucially, LD-FPG achieves this realism while also delivering **superior local geometry**. Its backbone dihedral JSD of **0.007** is **3-5x better** than all baselines, proving our model generates a flexible, all-atom ensemble with state-of-the-art accuracy.
>
> **Generalization:** To address your valid concern about applicability, the appendix will now present results from our broader validation studies. These studies, conducted during our initial model development to confirm the framework's robustness, show that our model consistently generates high-fidelity ensembles across a diverse set of targets, including:
> * **Other Class A GPCRs** (D1, A2A, $\beta$1-adrenergic).
> * **Diverse Folds from ATLAS**, including large systems like cytochrome P450 (~500 residues) and different architectures ($\alpha$-helical, $\beta$-sheet, mixed $\alpha/\beta$).
> * **Folding Dynamics**, demonstrated by modeling the full folding trajectory of the TRP-cage miniprotein.
>
> All-atom lDDT scores across this set are in the range of 0.68–0.78, with total JS divergences between 0.007 and 0.014. Simulation data for the additional GPCRs is now publicly available via our Zenodo link; a comprehensive list of all validated ATLAS structures will be included in the appendix.
>
>
> ## Questions
>
> ### 2.1 Loss/Architecture Definitions
>
> As addressed above (1.1), all key concepts will be clearly defined and cross-referenced within the main text.
>
> ### 2.2 Comparison with Existing Ensemble Generation Models
>
> As we detailed in our response above (Point 1.2), we have performed an extensive new benchmarking study against AlphaFlow, BioEmu, and Boltz-2. Regarding MD-Gen:  We chose not to include it in our benchmark since it is designed for trajectory generation  and is currently tuned for small peptides (≤50 residues) in solvent whereas our work focuses on sampling conformations from the equilibrium ensemble. Applying it to a 273-residue membrane protein like D2R is outside its intended domain and would not produce a fair comparison.
>
> ### 2.3 Latent Space Diffusion
>
> Thank you for this question. It highlights a critical design choice. We perform diffusion in latent space for three key reasons:
>
> 1. **Dynamics-Focused Learning:** By encoding only the deformations in a pooled latent space (relative to a static reference structure), the diffusion model can focus on learning the protein's internal, functionally relevant motions. The static fold and global orientation are factored out, allowing the model to learn a more targeted and meaningful distribution.
> 2. **Numerical Tractability and Stability:** The dimensionality is drastically reduced from ~6,600 degrees of freedom in Cartesian space for the D2R protein to ~100 dimensions in our pooled latent space. This makes the optimization stable and efficient. In our ablation studies, an equivalent DDPM trained directly in coordinate-space failed to converge entirely.
> 3.  **Implicit Physical Priors:** By learning to perturb a physically-correct reference structure, the decoder inherently preserves correct local bond geometry and chirality, improving realism. This is supported by our low side-chain dihedral JSDs.
>
> Our approach is consistent with strategies used in recent generative models. Given the challenges of all-atom diffusion, models like AlphaFold 3 [1] and Boltz-2 [2] use all-atom attention for detailed features, but then pool these representations to the residue level before applying diffusion in a lower-dimensional space—much like our residue-based pooling. Similarly, La-Proteina [3] keeps backbone coordinates explicit and diffuses side-chain information in a separate latent space, which resembles our sequential pooling setup. This confirms that our design addresses a widely recognized challenge in the field using practical, effective methods.
>
>
> [1] Abramson et al., Nature 2024 (AlphaFold 3)
> [2] Wohlwend et al., bioRxiv 2025 (Boltz-2)
> [3] Geffner et al., NVIDIA Research 2025 (La-Proteina)
>
>
> ---
>
> ### 2.4 D2R as a GPCR and Biological Relevance
>
> You are right that we should have made the GPCR context and its biological relevance more explicit. To address this directly, we are strengthening the manuscript with new, targeted analyses and revising the text for clarity.
>
> First, to validate that our model captures functionally critical motions, we performed a new analysis on two canonical GPCR activation metrics: the **TM6 outward movement** ($R^{3.50}-E^{6.30}$ distance) and the **NPxxY hinge twist** ($D^{2.50}-Y^{7.53}$ distance). Our results confirm that LD-FPG accurately reproduces the free-energy profiles of these key transitions.
>
> Furthermore, a key part of our original model validation involved testing its ability to learn from complex, non-equilibrium data. We trained LD-FPG on a **metadynamics trajectory** (~500ns) that explicitly samples the rare intermediate conformations along the D2R activation pathway. The model successfully learned from this sparse data and can generate realistic conformations that trace the entire transition path. This provides compelling evidence that the framework learns the underlying energy landscape, not just the dense regions of an equilibrium simulation.
>
> Both of these new analyses will be detailed in a **new Appendix G**. Finally, we will revise the **Introduction**, **Related Work**, and **figure captions** to consistently highlight the GPCR context.
>
> ---
>
> ### 2.4 Clash Counts
>
> Your concern is valid. Our response is two-fold:
>
> 1.  **The Correct Baseline for Clashes is the MD Ensemble:** There is no universal threshold for "acceptable" clashes. Standard validation tools like MolProbity are calibrated for static, minimum-energy crystal structures and are ill-suited for evaluating dynamic MD ensembles, which naturally explore higher-energy states and exhibit transient van der Waals violations. Therefore, the most relevant benchmark is the reference MD simulation itself. As we report in the manuscript (Line 300), the **ground truth MD ensemble has a non-zero average clash score of ≈1023**. Our best-performing models (e.g., Residue Pooling at 1145 clashes) are remarkably close to this physical reference, indicating a high degree of realism.
>
> 2.  **An Area for Future Work:** Managing steric clashes is a challenge for all generative models. We acknowledge the moderately higher clash count is a trade-off of our current approach. We are actively exploring physics-informed strategies, similar to those in models like Boltz-1, to further refine physical realism in future work. We will clarify this context in the revised manuscript.
>
>
> ### 2.6 Appendix Reference Typo
>
> The "Appendix ??" typo in Algorithm 1 will be corrected in the final manuscript.

---

> > ### Author Response · Authors · 2025-08-06
> > **Follow-up on Rebuttal for Submission 28539**
> >
> > Dear Reviewer ZSKy,
> >
> > As the discussion period is coming to a close, we wanted to briefly follow up on our rebuttal from July 30th. We were particularly keen to ensure you saw the new analysis and clarifications we provided in response to your thorough review.
> >
> > Our rebuttal now includes:
> > 1.  An extensive **benchmark against state-of-the-art models** like AlphaFlow and BioEmu, which was a primary concern you raised.
> > 2.  A summary of our **broader validation studies** to address your valid questions about generalization.
> > 3.  A detailed rationale for our use of **latent space diffusion** and further context on the **steric clash** evaluation.
> >
> > We believe these additions substantially strengthen the paper by addressing the main weaknesses you identified. We would be very grateful for any feedback, and we are on standby to answer any final questions you may have.
> >
> > Thank you for your time and consideration.

---

> > > ### Comment · Reviewer_ZSKy · 2025-08-06
> > >
> > > Thank you for the detailed explanation. The additional clarifications have helped highlight the significance of the work more clearly. I have accordingly raised my score.

---

> > > > ### Author Response · Authors · 2025-08-06
> > > >
> > > > Dear Reviewer ZSKy,
> > > >
> > > > Thank you so much for your positive follow-up and for raising your score. We are delighted to hear that our clarifications helped better convey the work's significance.
> > > >
> > > > Your feedback has been invaluable in improving the manuscript, and we truly appreciate your time and reconsideration.
> > > >
> > > > Thank you again.

---

### Note · Authors · 2025-08-12

Dear Area Chair and Reviewers,

Thank you for the constructive discussion that has significantly strengthened our work. The camera-ready manuscript will fully integrate the improvements developed during rebuttal.

We will integrate our new SOTA benchmark into the main text, showing LD-FPG matches MD-level flexibility (RMSF) while achieving superior local geometry (JSD). The appendix will be expanded with a GNN encoder comparison, broader validation on diverse proteins, and free-energy profiles for key GPCR activation metrics.

Fundamentally, LD-FPG is an ideological departure. By reframing the generative task from predicting absolute coordinates to learning a **low-dimensional latent space of internal deformations (Δz)**, we make all-atom ensemble generation stable and tractable. This intellectual shift allows a simple yet powerful architectural stack—like a ChebNet encoder and an MLP-based denoiser—to succeed, as the complexity is managed by the problem formulation itself rather than requiring more intricate equivariant or attention-based networks.

This practical approach also makes our framework a powerful add-on for bringing dynamics and all-atom detail to leading predictors. For example:

* **Animating AlphaFold 3:** **An AF3 structure can serve as a high-quality physical reference ($R_{ref}$)**. A lightweight LD-FPG model can then be trained on system-specific MD data to learn and sample a latent distribution of all-atom *deformations* around this static anchor, combining AF3's accuracy with realistic, learned flexibility.

* **Upgrading BioEmu:** **Our framework can serve as a powerful all-atom refinement layer for coarse-grained models like BioEmu.** The logic mirrors our Sequential Pooling architecture: for each Cα-ensemble frame generated by BioEmu, our decoder—trained on the corresponding *full-atom* source trajectories—conditionally generates the complete, physically realistic side-chain structure. This synergistically combines BioEmu's sampling of large-scale backbone motions with our model's ability to capture accurate, all-atom local chemistry.

In sum, our work provides both a powerful stand-alone tool and a set of modular recipes to endow existing SOTA models with realistic, all-atom dynamics. We are confident the final paper will be a clear, reproducible, and valuable resource for the community.

---

### Decision · Program_Chairs · 2025-09-17

**Decision:**

Accept (poster)

**Comment:**

This submission introduces LD-FPG (latent diffusion for full protein generation) for generating all-atom protein conformational ensembles by fusing GNNs and latent diffusion. It uses ChebNet to encode protein conformations into low-dimensional latents, employs three pooling strategies (blind, sequential, residue-based) to balance global flexibility and local precision, and trains a diffusion model on these latents to generate new conformations followed by decoding back to Cartesian coordinates with optional dihedral regularization.

Reviewers uniformly recognized LD-FPG’s value in addressing a critical gap: generating dynamic, full-atom ensembles (a rarity compared to static/backbone-only models) and its extensive and comphrensive experimental design. Authors addressed all critical reviewer concerns during rebuttal, with most reviewers upgrading their assessments or confirming concerns were resolved. Given the consensus that LD-FPG’s core concerns were resolved, its methodological and empirical strengths, and its impact on protein modeling, I recommend accepting this submission. I encourage the authors to follow through on their commitments in the final manuscript to further strengthen the work.